# Seeing Realism from Simulation: Efficient Video Transfer for Vision-Language-Action Data Augmentation

**Chenyu Hui** [1 2 *]  **Xiaodi Huang** [1 3 *]  **Siyu Xu** [4]  **Yunke Wang** [4]  **Shan You** [5]  **Fei Wang** [6]  **Tao Huang** [1 †]  **Chang Xu** [4]

## Abstract

Vision-language-action (VLA) models typically rely on large-scale real-world videos, whereas simulated data, despite being inexpensive and highly parallelizable to collect, often suffers from a substantial visual domain gap and limited environmental diversity, resulting in weak real-world generalization. We present an efficient video augmentation framework that converts simulated VLA videos into realistic training videos while preserving task semantics and action trajectories. Our pipeline extracts structured conditions from simulation via video semantic segmentation and video captioning, rewrites captions to diversify environments, and uses a conditional video transfer model to synthesize realistic videos. To make augmentation practical at scale, we introduce a diffusion feature-reuse mechanism that reuses video tokens across adjacent timesteps to accelerate generation, and a coreset sampling strategy that identifies a compact, non-redundant subset for augmentation under limited computation. Extensive experiments on Robotwin 2.0, LIBERO, LIBERO-Plus, and a real robotic platform demonstrate consistent improvements. For example, our method improves RDT-1B by 8% on Robotwin 2.0, and boosts $\pi_0$ by 5.1% on the more challenging LIBERO-Plus benchmark. Code is available at: CODE.

## 1. Introduction

The advent of large-scale Vision-Language-Action (VLA) models has marked a significant milestone in robotics (Liu et al., 2024; Black et al., 2024; Intelligence et al., 2025b;a; Liu et al., 2024; Zhao et al., 2023; Xu et al., 2025a), enabling robots to interpret natural language instructions and execute complex manipulation tasks. These models typically rely on extensive datasets of real-world robotic trajectories to learn generalizable policies. However, collecting such real-world data is inherently costly, time-consuming, and difficult to scale, posing a major bottleneck to broader development and deployment. In contrast, simulated data offers a highly inexpensive alternative. But as noted in recent studies, models that achieve near-perfect performance within a specific simulation benchmark often fail catastrophically when faced with minor perturbations in object layout, lighting, camera viewpoint, or instruction phrasing in scenarios with disturbances (Fei et al., 2025; Zhou et al., 2025b; Xu et al., 2025b; Pei et al., 2025), revealing that many models merely memorize training sequences rather than learning robust, semantic task understanding.

We address the critical lack of visual and environmental diversity in robotic training data by proposing an efficient, scalable video augmentation framework. Our approach transforms source video sequences into visually diverse training videos while strictly preserving underlying task semantics and action trajectories. The pipeline first extracts structured conditions from simulation, diversifies the environmental context, and then synthesizes high-fidelity videos using a conditional video diffusion model. To mitigate the substantial computational cost of video transfer, we propose accelerating the diffusion generation by caching velocity during the denoising process. We design adaptive strategies that optimize the trade-off between quality and efficiency. Consequently, this velocity-caching mechanism achieves an over 60% reduction in generation time while maintaining high model accuracy.

Furthermore, a coreset sampling method is designed to achieve efficient scalability by only augmenting a subset of the large-scale dataset. For the sake of sampling diverse trajectories that maximize the performance, we build a graph structure that balances the difficulty and diversity among data nodes, where the difficulty and diversity are estimated by a video embedding model and a pretrained VLA policy. By prioritizing samples that are both challenging and highly

---

*Equal contribution. Work was done during internship at SJTU. [1]Shanghai Jiao Tong University [2]Xi'an Jiaotong University [3]Institute of Automation, Chinese Academy of Sciences [4]The University of Sydney [5]SenseTime Research [6]University of Science and Technology of China. Correspondence to: Tao Huang <t.huang@sjtu.edu.cn>.

*Proceedings of the 43rd International Conference on Machine Learning*, Seoul, South Korea. PMLR 306, 2026. Copyright 2026 by the author(s).

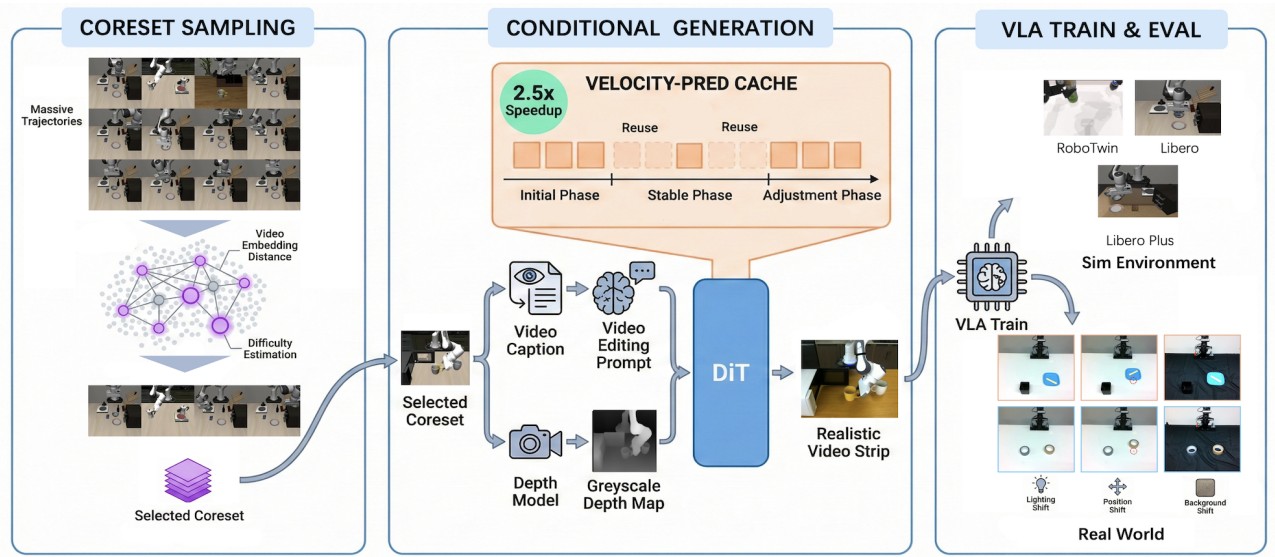

*Figure 1.* Overall framework of the proposed method. Given a large-scale simulation training set, we introduce a coreset sampling algorithm to select important and diverse samples, which are then augmented to realistic video strips and used for training.

diverse, we ensure the significance of the augmentation effect in large-scale datasets.

We conduct extensive experiments across multiple benchmarks, including Robotwin 2.0 (Chen et al., 2025), LIBERO (Liu et al., 2023), and the more challenging LIBERO-Plus (Fei et al., 2025) to demonstrate the effectiveness of our framework. We also conduct real-world robotic experiments.

In summary, our contributions are threefold:

- We present an end-to-end video transfer named efficient transfer framework that effectively bridges the sim-to-real gap for VLA data augmentation.

- We design a segmented (three-stage) velocity caching strategy, which better matches the dynamics of conditional video generation. It is proven that adopting this strategy can save over 60% transferring time.

- We propose a trajectory-level coreset formulation that combines policy difficulty (measured by the policy loss of RDT-1B), and visual diversity (measured via Cosmos-Embed1 representations), leading to more effective selection of high-value training trajectories.

## 2. Related Work

### 2.1. Vision-Language-Action Models

Since the development of VLM and its been pivotal in advancing robotic control by providing rich multi-modal representations. Recent VLA models are finetuned on large-scale robotic trajectory data in an end-to-end mode to ob-

tain more generalizable policies (Brohan et al., 2022; Ebert et al., 2021). Widely used OpenVLA (Kim et al., 2024) is based on prismatic (Karamcheti et al., 2024). RDT combines the advantage of diffusion model and transformer (Liu et al., 2024). ACT (Zhao et al., 2023) introduced chunking for actions based on transformer. The OpenPi series like $\pi_0$ (Black et al., 2024), $\pi_{0.5}$ (Intelligence et al., 2025b) and the newest $\pi_{0.6}$ (Intelligence et al., 2025a) gains significant performance superiority by enhancing existing VLMs with a dual-system VLA architecture, retaining the text-generation ability of the VLM, and combining with reinforcement learning (Li et al., 2026b), respectively.

### 2.2. Video Generation

Recent advances in diffusion-based video generation and style transfer have established this as a pivotal area in multimodal modeling, building upon foundational works like Stable Diffusion (Rombach et al., 2022), Stable Video Diffusion (Blattmann et al., 2023). Contemporary research has expanded these frontiers: Sora incorporates a temporal VAE and Transformer backbone; Wan (Wan et al., 2025) and VACE (Jiang et al., 2025) employ mixture-of-experts architectures; while NVIDIA's Cosmos series (Alhaija et al., 2025), particularly Cosmos Transfer (Ali et al., 2025), provides a robust foundation for world model-based video transfer. Concurrently, generative techniques are increasingly applied to embodied data, with methods like Rebot (Fang et al., 2025b), EgoDemoGen (Xu et al., 2025c), EMMA (Dong et al., 2025), Embodied Dreamer (Wang et al., 2025), Gigaworld-0 (Team et al., 2025b), RoboTransfer (Liu et al., 2025b) and Gigabrain-0 (Team et al., 2025a) demonstrating effective real-to-sim and sim-to-real data generation.

## 2.3. Data Centric AI and Efficient Training

Data-centric approaches have proven crucial for enhancing model performance and training efficiency, evolving from early heuristic-based filtering methods like CCNet (Wenzek et al., 2020) and C4 (Raffel et al., 2020) to sophisticated techniques emphasizing deduplication (Lee et al., 2022) and semantic redundancy elimination (Abbas et al., 2023). Beyond quality filtering, data diversity plays an equally vital role in robustness, as demonstrated by The Pile's emphasis on heterogeneous data (Gao et al., 2020) and Falcon Series' success with web-scale curation (Penedo et al., 2023). Domain-specific selection methods like DSIR (Xie et al., 2023) leverage importance resampling for target distribution alignment, while theoretical advances (Sorscher et al., 2022) show intelligent pruning can achieve exponential efficiency gains. This paradigm culminates in $\mathbb{D}^2$ Pruning (Maharana et al., 2023), which advocates balanced coreset selection based on complementary diversity-difficulty criteria.

## 3. Method

We propose a VLA data augmentation framework to improve sim-to-real generalization. We first analyze the generalization failures of simulation-trained models, then introduce an end-to-end pipeline that generates realistic videos from semantic and structural cues. To scale augmentation efficiently, we develop a diffusion acceleration method based on velocity caching and a coreset sampling strategy for selective data generation.

### 3.1. Motivation: The Brittleness of VLA Models

VLA models trained on robotic manipulation data often achieve high in-distribution performance but fail under distribution shifts. The core issue is **limited visual and environmental diversity** in training data: simulated environments are overly clean and static, while real-world datasets cover only narrow conditions due to collection constraints. This lets models **overfit to spurious regularities** rather than learning robust instruction-to-action mappings.

Recent benchmarks quantify this brittleness: LIBERO-Plus (Fei et al., 2025) shows success rates dropping from 95% to under 30% with minor perturbations in object layout, camera angle, or lighting; LIBERO-PRO (Zhou et al., 2025b) reports near-zero accuracy when object positions and instructions are modified. As illustrated in Figure 2, baseline VLA models fail under such perturbations, revealing memorization of fixed action sequences rather than semantic task understanding.

The root cause lies in sterile training data that lacks the complexity of real-world contexts. Without diverse backgrounds, lighting, viewpoints, and object configurations, models latch onto superficial correlations (Xu et al., 2025b;

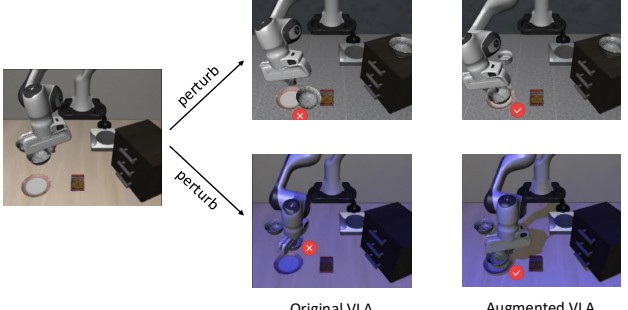

*Figure 2.* Examples from LIBERO-Plus evaluation. The baseline VLA model fails under environment perturbations such as texture change (upper) and lighting change (lower), while the model trained with our augmented data performs the tasks correctly, showing stronger generalization.

Fang et al., 2025a) and under-utilize language inputs (Fei et al., 2025; Zhou et al., 2025b; Li et al., 2026a). This motivates our approach: augmenting training distributions with visually varied yet semantically faithful data to enable robust generalization.

### 3.2. Diversifying Data via Conditional Video Transfer

We present a video augmentation framework that transforms source robotic videos into visually diverse counterparts while preserving task semantics and action fidelity. As illustrated in Figure 1, our pipeline consists of four stages: caption generation, caption rewriting, structured condition extraction, and conditional video synthesis.

**Semantic and Structural Condition Extraction.** We first extract descriptive captions from source videos using a temporal video captioning model (*i.e.*, VideoChat2 (Li et al., 2025)), which summarizes interactions, objects, and spatial relations. These captions are then rewritten by a large language model (*i.e.*, Qwen3-8B (Yang et al., 2025)) to introduce environmental variations such as background and object color changes, yielding diverse scene contexts while preserving task intent (details in B.2). To maintain geometric consistency, we also extract depth maps from source videos as structural control signals, which provide stable, geometry-preserving guidance compared to alternatives like blur, edge, or segmentation (Ravi et al., 2024).

**Conditional Video Synthesis.** The final stage employs a conditional video diffusion model (*i.e.*, Cosmos-Transfer 2.5 (Ali et al., 2025)) to synthesize realistic videos conditioned on the rewritten caption and depth input. The model iteratively denoises video tokens, generating temporally coherent scenes that retain the original action trajectory while varying the visual context. This process enriches the training distribution with diverse visual conditions without requiring additional real-world data collection.

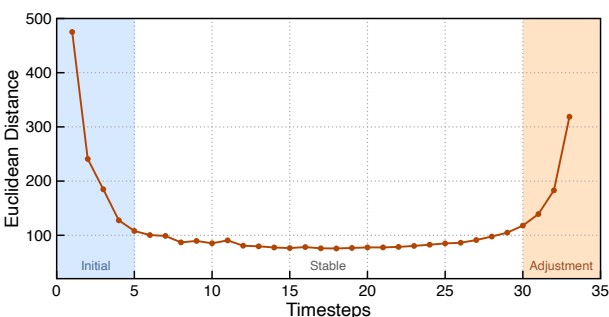

*Figure 3.* Euclidean distance between adjacent velocity predictions. A stable phase with minimal changes enables caching and reuse.

## 3.3. Efficient Video Generation via Velocity Caching

Recent conditional video diffusion models such as Cosmos-Transfer (Ali et al., 2025) and Wan (Wan et al., 2025) achieve strong visual fidelity but suffer from prohibitively high inference costs[1], posing a major obstacle to large-scale data augmentation. We propose a velocity caching mechanism that exploits temporal redundancy in the diffusion process to significantly accelerate generation while preserving quality.

**Observation: Redundancy in Denoising.** In flow-based diffusion models (Lipman et al., 2022), each denoising step computes a velocity field $v_\theta(x_t, t)$ to update the latent:

$$x_{t+1} = x_t + \Delta t \cdot v_\theta(x_t, t). \tag{1}$$

This velocity prediction accounts for over 70% of per-step runtime due to the full forward pass through a large video diffusion transformer. Inspired by recent caching strategies (Liu et al., 2025a; Zhou et al., 2025a; Li et al., 2023), we analyze the temporal dynamics of velocity predictions during denoising.

As shown in Figure 3, measuring the Euclidean distance $\|v_{t+1} - v_t\|$ across timesteps reveals three distinct phases: (1) an *initial phase* with rapidly changing predictions, (2) a *stable phase* with minimal variation, and (3) a *final adjustment phase* for fine-grained refinement. This suggests that we can avoid re-computing $v_\theta$ at every step during the stable phase by reusing previously cached values.

**Three-Stage Caching Strategy.** Based on this observation, we divide the $N$-step denoising into three stages: (1) **Initial Phase** ($t < t_s$): velocity predictions are computed at each step due to high variability; (2) **Stable Phase** ($t_s \le t < t_f$): velocities are computed every $\alpha$ steps and cached values are reused in between; (3) **Adjustment Phase** ($t \ge t_f$): full computation resumes for final refinement. We detect the

---

[1]Cosmos-Transfer 1 takes ~40 minutes for a 5-second 720p video on one A100 GPU.

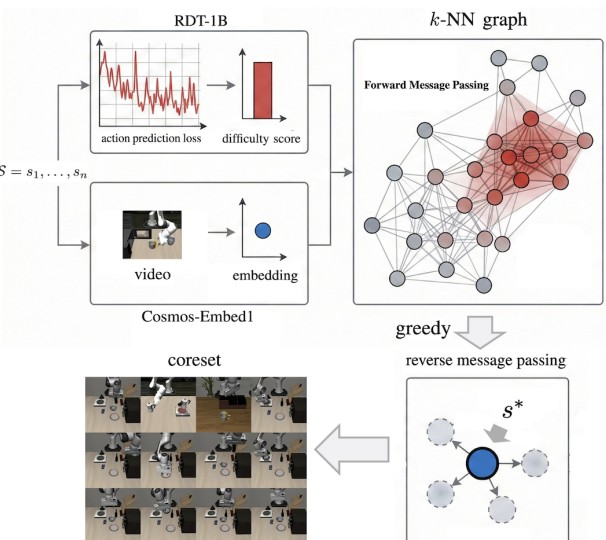

*Figure 4.* The proposed VLA coreset sampling algorithm.

stable phase onset using a relative smoothness threshold:

$$\frac{\|v_t - v_{t+1}\|}{\|v_0 - v_1\|} < k. \tag{2}$$

With $k = 0.4$, $\alpha = 8$, and $m = 3$ adjustment steps, this strategy achieves an average **61.2%** time reduction on RoboTwin 2.0 with negligible quality degradation, enabling efficient scaling of our augmentation pipeline.

## 3.4. Selective Augmentation via Coreset Sampling

Although our video augmentation framework enhances VLA training, applying it to all trajectories in a large-scale dataset is computationally prohibitive. To maximize augmentation utility under limited resources, we propose a coreset sampling strategy that identifies trajectories which are both challenging for policy models and visually diverse, as illustrated in Figure 4. Our method extends $\mathbb{D}^2$ Pruning (Maharana et al., 2023) to embodied video data.

Given a set of simulated trajectories $\mathcal{S} = \{s_1, \ldots, s_n\}$, our objective is to select a compact subset $\mathcal{S}' \subset \mathcal{S}$ that maximizes downstream benefit for sim-to-real transfer.

**Difficulty Estimation via Policy Loss** We quantify trajectory difficulty using a pre-trained VLA policy, such as RDT-1B (Liu et al., 2024). For each trajectory $s_i$, we compute the average action prediction loss over a sampled subset of time steps $\mathcal{T}_i$ (to reduce computational cost):

$$x_i = \frac{1}{|\mathcal{T}_i|} \sum_{t \in \mathcal{T}_i} \mathcal{L}_{\text{policy}}(s_i^{(t)}; \theta). \tag{3}$$

A higher $x_i$ indicates that the policy finds the trajectory more challenging, which typically corresponds to edge cases or difficult configurations for manipulation.

**Diversity Estimation via Visual Embedding Topology**
To capture visual and semantic diversity, we extract trajectory embeddings and define neighbors using a state-of-the-art video representation model, Cosmos-Embed1 (NVIDIA et al., 2025), denoted $\phi(s_i) \in \mathbb{R}^{768}$. We build a sparse $k$-NN graph over these embeddings with nodes $v_i = \phi(s_i)$ and edges weighted by an RBF kernel:

$$e_{i,j} = \exp(-\gamma_f \cdot \|v_i - v_j\|^2). \tag{4}$$

This graph captures the topological structure of the visual space, enabling both forward and reverse message passing to score and prune redundant samples.

**Scoring via Forward Message Passing** To prioritize clusters of difficult and non-redundant samples, we compute an updated difficulty score $x_i'$ that aggregates neighborhood difficulty:

$$x_i' = x_i + \sum_{j \in \mathcal{N}(i)} e_{i,j} \cdot x_j. \tag{5}$$

This promotes samples that are not only hard in isolation but are also surrounded by similarly hard cases, emphasizing policy failure regions in the data.

**Coreset Construction with Redundancy Suppression** We perform greedy selection using $x_i'$ scores and penalize neighbors of selected samples to avoid redundancy:

1. Select the trajectory with the highest score $s^* = \arg\max_{i \in \mathcal{S}_{\text{remaining}}} x_i'$, and add $s^*$ to selected coreset $\mathcal{S}'$.

2. Suppress scores of visually similar neighbors using reverse message passing:

$$x_j' \leftarrow x_j' - \exp(-\gamma_r \cdot \|v_{s^*} - v_j\|^2) \cdot x_{s^*}',$$

where $\gamma_r$ controls the suppression radius.

This iterative procedure balances sample hardness and coverage, yielding trajectories that are visually diverse and strategically valuable for sim-to-real transfer. By targeting augmentation toward this curated coreset, we achieve computational savings while preserving augmentation benefits. The details of hyperparameters above are as follows: $k = 5, \gamma_r = 0.1, \gamma_f = 0.1$.

# 4. Experiments

In the experiments below, first we prove the effectiveness of our augmentation method on single-task from Robotwin 2.0. Then, we move on to large-scale learning using a coreset sampling policy. To verify the advantages of our method under environment with specific disturbances, we evaluate on LIBERO-Plus. Finally, to study the validity of our method in a real-world experiment, we design tasks and evaluate with an AgileX Piper robot arm under distributed environments, as shown in Figure 5.

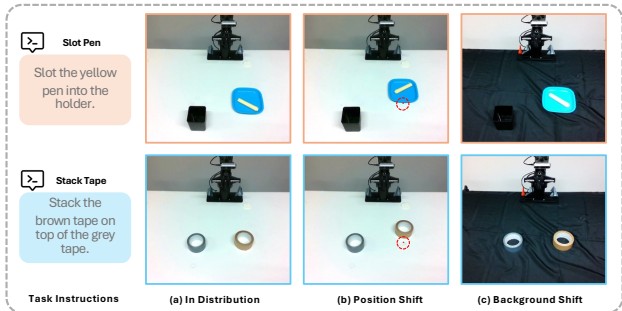

*Figure 5.* Two manipulation tasks (Slot Pen and Stack Tape) under three test conditions: (a) In-Distribution, (b) Position Shift, and (c) Background Shift.

*Table 1.* Performance (%) comparison of Hard and Easy scenarios of Robotwin 2.0 (Chen et al., 2025) - single-task learning. We train RDT (Liu et al., 2024) with the original simulation data (Ori.) *vs.* augmented data (Aug.).

| Task Name | Hard | | Easy | |
|---|---|---|---|---|
| | Ori. | Aug. | Ori. | Aug. |
| adjust_bottle | 72.0 | 82.0 (+10.0) | 74.0 | 90.0 (+16.0) |
| beat_block_hammer | 36.0 | 48.0 (+12.0) | 76.0 | 84.0 (+12.0) |
| pick_dual_bottles | 14.0 | 20.0 (+6.0) | 40.0 | 36.0 (-4.0) |
| place_burger_fries | 26.0 | 38.0 (+12.0) | 46.0 | 54.0 (+8.0) |
| open_laptop | 30.0 | 44.0 (+14.0) | 56.0 | 56.0 |
| move_can_pot | 12.0 | 18.0 (+6.0) | 26.0 | 34.0 (+8.0) |
| rotate_qrcode | 4.0 | 8.0 (+4.0) | 4.0 | 10.0 (+6.0) |
| grab_roller | 40.0 | 50.0 (+10.0) | 72.0 | 74.0 (+2.0) |
| average | 29.0 | 39.0 (+10.0) | 49.0 | 55.0 (+6.0) |

## 4.1. Experimental Results on Robotwin 2.0

Following the experimental framework established by Robotwin 2.0 (Chen et al., 2025), this study utilizes the Aloha AgileX dual-arm robot system. For the strategy model selection, we adopted both RDT-1B (Liu et al., 2024) ACT (Zhao et al., 2023) and $\pi_0$ (Black et al., 2024) architectures, both fine-tuned based on officially pre-trained weights.

**(1) Experiments on single-task learning.** We use 50 expert demonstration trajectories in a clean environment to train 10 tasks independently as the original baseline for single-task learning. For our method, we extend augmented first-person perspective video data of all the trajectories, then train the same model for the same steps. We conducted 50 tests on each task under both "Easy" and "Hard" settings. The "Easy" mode corresponds to a *clean* environment, while the "Hard" mode introduces *domain-randomized* factors, including clutter, lighting, textures, and height variations.

**Results.** Table 1 reports the performance of RDT-1B. Compared to the original baseline, our method yields substantial

gains, most notably a 10% improvement in "Hard", which includes domain-randomized factors and better reflects real-world complexity. Even in the clean and easy scenario, we observe a 6% increase, highlighting our method's ability to enhance generalization and mitigate overfitting.

Additional results in the Appendix further validate our approach. We report an average increase of 2.0 in $\pi_0$ (Table 9) and 3.0 in ACT (Table 8) under the hard setting of Robotwin 2.0 single-task learning, confirming the robustness of our augmentation strategy across different VLA policies.

**(2) Experiments on multi-task learning**. We expanded the experimental scope to 32 diverse robot operation tasks with 300 trajectories on each task, resulting in a larger amount of 9600 training trajectories. For our method, we adopt a coreset sampling algorithm to select 10% key samples in the full set, resulting in 9600 training samples with 90% original and 10% augmented. When training, both experiments utilize the official pre-trained weights of RDT-1B and undergo large-scale multi-task joint training (100k steps). For evaluation metrics, we evaluated nine representative tasks for 50 times in "Hard" scenario.

*Table 2.* Performance (%) comparison of Hard scenarios of Robotwin 2.0 (Chen et al., 2025) - multi-task learning. We train RDT (Liu et al., 2024) with the original simulation data (Ori.) *vs.* augmented data (Aug.).

| Task Name | Ori. | Aug. |
|---|---|---|
| adjust_bottle | 20.0 | 56.0 (+36.0) |
| beat_block_hammer | 22.0 | 30.0 (+8.0) |
| place_burger_fries | 22.0 | 18.0 (-4.0) |
| pick_dual_bottles | 6.0 | 10.0 (+4.0) |
| place_object_basket | 6.0 | 14.0 (+8.0) |
| move_can_pot | 8.0 | 16.0 (+8.0) |
| stack_bowls_three | 20.0 | 22.0 (+2.0) |
| click_alarmclock | 48.0 | 52.0 (+4.0) |
| move_playingcard_away | 16.0 | 22.0 (+6.0) |
| open_laptop | 60.0 | 66.0 (+6.0) |
| average | 23.0 | 31.0 (+8.0) |

**Results**. As shown in Table 2, despite having 300 training trajectories per task, the baseline model struggles to generalize in the highly repetitive and visually uniform Robotwin environment. In contrast, augmenting just 10% of the data with our method yields a significant performance boost. This demonstrates that our approach provides valuable visual and semantic diversity that the original dataset lacks, effectively regularizing the model and enhancing generalization, even under strong simulation constraints.

### 4.2. Experiments on LIEBRO-Plus and LIBERO

To validate the effectiveness and generality of our proposed method on widely accepted public benchmarks, we selected LIBERO (Liu et al., 2023), and LIBERO-Plus (Fei et al., 2025) to conduct experiments using $\pi_0$ (Black et al., 2024) and $\pi_{0.5}$ (Intelligence et al., 2025b).

We train all the models using the default standard configuration of OpenPi [2]. For the baseline, we finetune the pretrained models with the original dataset. While for our method, we augment a coreset of 50% samples under two data compositions: (1) mixture strategy for mixing the augmented samples and all the original samples together; (2) replacement strategy for replacing the selected coreset with augmented ones.

*Table 3.* Performance (%) comparison of spatial suite of LIBERO-Plus (Fei et al., 2025) using $\pi_0$ and $\pi_{0.5}$.

| Perturbation Types | $\pi_0$ | | $\pi_{0.5}$ | |
|---|---|---|---|---|
| | Ori. | Aug. | Ori. | Aug. |
| light conditions | 75.0 | 78.7 (+3.7) | 94.5 | 97.9 (+3.2) |
| objects layout | 69.6 | 86.2 (+16.6) | 97.9 | 97.4 (-0.5) |
| background textures | 81.1 | 87.6 (+6.5) | 95.7 | 95.3 (-0.4) |
| sensor noise | 19.9 | 18.2 (-1.7) | 91.2 | 93.4 (+2.2) |
| language instructions | 37.9 | 55.9 (+22.0) | 90.0 | 90.0 |
| robot initial states | 10.3 | 6.3 (-4.0) | 82.6 | 84.9 (+2.3) |
| camera view points | 21.3 | 15.2 (-6.1) | 79.3 | 79.5 (+0.2) |
| average | 42.7 | 47.8 (+5.1) | 89.8 | 90.8 (+1.0) |

**Results**. After 30k-step training, we evaluate on both $\pi_0$ (mixture strategy) and $\pi_{0.5}$ (replacement strategy) on the spatial suite of LIEBRO-plus, which contains **2402** evaluation settings. According to the results in Table 3, especially on $\pi_0$ 's performance comparison, it shows that adopting our augmentation method along with a training strategy can enhance models' ability in simulated environments with disturbances like changing light conditions. The slight drop in camera viewpoints and robot initial states stems from a limitation of the current method: it mainly augments appearance, while these perturbations are geometry- and viewpoint-dependent. The results on $pi_{0.5}$ also denote that even with a high-performance model, using our method can still lead to an obvious increase in some disturbance dimensions like sensor noise, light conditions, and robot initial states.

Additionally, we also evaluate on LIBERO and observe a slight performance drop when training with augmented data: $\pi_0$ decreases by an average of 0.2 and $\pi_{0.5}$ by 0.5 across four task suites. This is expected, as LIBERO's evaluation settings are nearly identical to its original training distribution—making aggressive augmentation introduce distributional disturbances rather than helpful diversity. Full results are provided in Table 10 (Appendix).

---

[2]https://github.com/Physical-Intelligence/openpi

We further compare mixture *vs.* replacement strategies for incorporating augmented data. Interestingly, $\pi_0$ performs best under the mixture strategy, while $\pi_{0.5}$ sees greater gains with replacement. This suggests that stronger pretrained models like $\pi_{0.5}$ benefit more from challenging augmentations that shift the training distribution, while weaker models prefer a balance between clean and augmented samples. See Table 11 in the Appendix for details.

### 4.3. Real-World Experiments and Results

To validate the effectiveness of our video augmentation framework in bridging the sim-to-real gap, we conduct comprehensive experiments on a physical robotic platform with diverse out-of-distribution (OOD) test scenarios.

**Experimental setup.** We deploy our method on an AgileX Piper manipulator using two state-of-the-art VLA model variants: $\pi_0$ (Black et al., 2024) and $\pi_{0.5}$ (Intelligence et al., 2025b). We evaluate on two representative manipulation tasks: (1) *Stack Tape*: "stacking a brown tape roll on top of another brown tape"; and (2) *Slot Pen*: "inserting a yellow pen into a holder". To systematically evaluate generalization capability, we design three test conditions with increasing levels of distribution shift, namely In-Distribution, Position Shift (OOD), and Background Shift (OOD). For details, see Figure 5 and Section B.4 in Appendix.

**Results.** Table 4 presents the real-world experimental results across all tasks and test conditions. Our video augmentation method demonstrates consistent and substantial improvements over the baselines across both models.

*Table 4.* Real-world results showing success counts (out of 10 trials) under three test conditions.

| Method | In-Dist. | | Position | | Background | | Avg. |
|---|---|---|---|---|---|---|---|
| | Slot. | Stack. | Slot. | Stack. | Slot. | Stack. | |
| $\pi_{0.5}$ | 9/10 | 10/10 | 5/10 | 4/10 | 4/10 | 4/10 | 60% |
| + Ours | **10/10** | **10/10** | **7/10** | **6/10** | **6/10** | **5/10** | **73%** |
| $\pi_0$ | 9/10 | 9/10 | 3/10 | 5/10 | 6/10 | 4/10 | 60% |
| + Ours | **10/10** | **10/10** | **5/10** | **8/10** | **5/10** | **7/10** | **75%** |

**Analysis.** Several key observations emerge:

*(1) Consistent improvements across VLA architectures.* Our model-agnostic method yields 13% and 15% absolute gains in average success rates for $\pi_{0.5}$ and $\pi_0$, respectively. This validates that the visual diversity from our video transfer pipeline benefits various VLA backbones without requiring architecture-specific modifications.

*(2) Enhanced robustness to spatial perturbations.* Significant gains under position shifts (e.g., $\pi_0$ on Stack Tape: 5/10 $\rightarrow$ 8/10) suggest the model learns generalizable spatial rea-

soning rather than memorizing fixed object configurations or action sequences.

*(3) Improved visual domain generalization.* Improved performance under background shifts (e.g., $\pi_0$ on Stack Tape: 4/10 $\rightarrow$ 7/10) indicates that our synthetic diversity fosters robust feature representations that are resilient to task-irrelevant visual changes.

*(4) Maintained in-distribution performance.* Augmented models maintain near-perfect in-distribution success rates (9–10/10), confirming that our pipeline preserves task-relevant semantics and action fidelity while introducing beneficial diversity.

### 4.4. Ablation Study

**Video augmentation with or without velocity cache.** To find the influence of using cache-based acceleration on augmented videos, we train RDT-1B on four tasks from Robotwin 2.0 - single-task learning with or without velocity cache. As the results summarized in Table 5, both cases obtain similar accuracies and achieve significant improvements over the original baseline. We also compare their visual quality in Figure 13 and Figure 14 of Appendix. Although the cache acceleration technique introduces minor degradation in texture details at the pixel level, our method successfully preserves environmental semantic information, which is essential for downstream policy learning, significantly reducing generation overhead without compromising the effectiveness of the augmented data.

*Table 5.* Comparison of RDT-1B model finetuned using augmented data with or without velocity cache acceleration on Robotwin 2.0.

| Task Name | Hard | | Easy | |
|---|---|---|---|---|
| | w/ acc. | w/o acc. | w/ acc. | w/o acc. |
| move_can_pot | 16.0 | 18.0 | 34.0 | 34.0 |
| open_laptop | 42.0 | 44.0 | 54.0 | 56.0 |
| rotate_qrcode | 10.0 | 8.0 | 8.0 | 10.0 |
| place_burger_fries | 38.0 | 38.0 | 54.0 | 52.0 |
| average | 26.5 | 27.0 | 37.5 | 38.0 |

**Evaluation of coreset sampling.** Figure 6 visualizes the video embeddings using t-SNE. The original dataset exhibits distinct clusters, each corresponding to semantically different tasks or visual contexts, with varying difficulty levels. Our selected coreset samples are well distributed across these clusters, covering both common and rare regions of the data manifold. Compared to random sampling, our method avoids over-representing redundant trajectories and instead targets high-loss, underrepresented areas. This balanced selection ensures that the augmented data introduces both semantic richness and policy-critical challenge, reinforcing

generalization while avoiding unnecessary redundancy.

To evaluate how the proportion of selected samples affects performance, we conduct an ablation using $\pi_0$ across 796 evaluation settings from LIBERO-Plus's spatial suite.

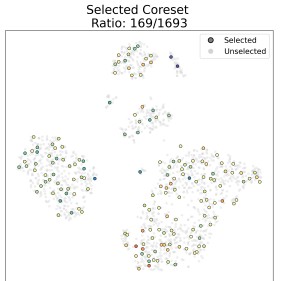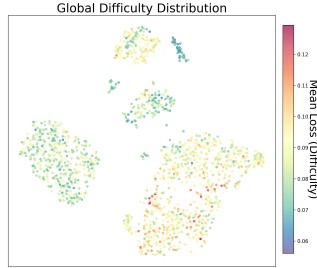

*Figure 6.* Visualization of coreset sampling on the LIBERO training dataset with a 10% sampling budget. Right: the global difficulty distribution, where the color spectrum represents the mean policy loss (redder colors indicate higher difficulty). Left: the selected coreset overlaid on the full dataset.

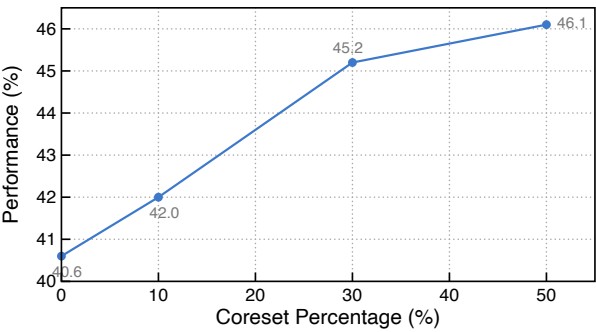

*Figure 7.* Performance comparison of different coreset sampling percentages on LIBERO-Plus spatial suite.

As shown in Figure 7, even a small augmentation ratio (10%) yields a substantial improvement, with performance continuing to rise as more samples are included. However, the gain between 30% and 50% is only 0.9%, indicating diminishing returns and highlighting the effectiveness of our coreset sampling in identifying critical and diverse samples.

### 4.5. Video Transferring Performance Assessment

To validate the efficacy of our proposed framework, we conducted a comparative study against the state-of-the-art baseline, RoboTransfer (Liu et al., 2025b). We evaluated performance across five representative tasks to assess both geometric consistency and semantic alignment. We employed two distinct sets of metrics to evaluate the synthesized video quality:

**Geometric Consistency**: We measured the depth fidelity between generated sequences and original sequences using Video Depth Anything (Yang et al., 2024). Metrics include

the Root Mean Square Error (RMSE), Absolute Relative Error (Abs.Rel), and Squared Relative Error (Sq.Rel). Lower values indicate higher geometric consistency.

**Semantic Alignment**: We utilized VideoCLIP-XL (Wang et al., 2024) to compute the prompt-video similarity, quantifying how well the generated video adheres to the provided textual instructions. Higher similarity scores indicate better semantic correspondence.

The quantitative results are summarized in Table 6 and Table 7. Our method consistently outperforms RoboTransfer across all evaluated tasks.

*Table 6.* RoboTransfer Results

| task | RMSE | Abs.Rel | Sq.Rel | sim | time |
|------|------|---------|--------|-----|------|
| adjust bottle | 0.46 | 0.37 | 0.39 | 21.5 | 340s |
| beat hammer | 0.30 | 0.28 | 0.16 | 22.3 | 368s |
| handover | 0.49 | 0.32 | 0.20 | 18.6 | 710s |
| hanging mug | 0.41 | 0.34 | 0.26 | 24.2 | 814s |
| pick bottles | 0.37 | 0.30 | 0.18 | 20.3 | 372s |

*Table 7.* Ours Results

| task | RMSE | Abs.Rel | Sq.Rel | sim | time |
|------|------|---------|--------|-----|------|
| adjust bottle | 0.28 | 0.16 | 0.07 | 26.3 | 441s |
| beat hammer | 0.18 | 0.14 | 0.03 | 25.6 | 453s |
| handover block | 0.23 | 0.17 | 0.06 | 27.7 | 619s |
| hanging mug | 0.13 | 0.12 | 0.02 | 26.4 | 783s |
| pick bottles | 0.12 | 0.11 | 0.03 | 26.8 | 448s |

Regarding geometric consistency, we observed a substantial reduction in error rates. Notably, the RMSE improved from 0.46 to 0.28 in the "adjust bottle" task with Abs.Rel and Sq.Rel metrics demonstrating a 2–6× reduction across the board. Furthermore, our approach achieved superior semantic alignment, with prompt similarity scores consistently increasing from the 18.6–24.2 range to 25.6–27.7, while our method incurs a marginal increase in average runtime.

## 5. Conclusion

We present a scalable video augmentation framework designed to address the generalization limitations of vision-language-action models trained on simulation. Our method introduces semantic and visual diversity into synthetic training data by combining structured caption rewriting, conditional video generation, efficient velocity-prediction caching, and coreset-based selective augmentation. Through experiments on Robotwin, LIBERO, and LIBERO-Plus, we demonstrate consistent performance improvements and enhanced robustness to real-world perturbations. Our results highlight that when diverse and targeted, even modest augmentation is effective in improving model generalization.

## Acknowledgements

This work was supported by the National Natural Science Foundation of China under Grant 62506235.

## Impact Statement

**Ethical Considerations** This research focuses on enhancing the generalization of Vision-Language-Action models through efficient sim-to-real video augmentation. While aiming to reduce reliance on costly real-world data, we emphasize the need for ethical vigilance in deploying such technologies. Key considerations include ensuring transparency in generated data, mitigating biases inherited from simulations or language models, and maintaining alignment with human values in autonomous decision-making. We encourage ongoing community efforts to address these ethical challenges alongside technical progress.

**Societal Implications** By lowering barriers to robust robotic system development, this work could positively impact fields like industrial automation and assisted services. However, broader adoption may also introduce societal shifts, such as changes in labor dynamics or unintended use cases. We advocate for proactive dialogue to guide responsible innovation, ensuring benefits are widely distributed while risks are carefully managed. In line with standard practice, we defer detailed societal impact discussions to broader interdisciplinary efforts.

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

# A. Additional Experiments

## A.1. Results of More Model Variants on RoboTwin 2.0

For ACT and $\pi_0$, we carry out an experiment on a single task from RoboTwin 2.0 using 50 trajectories collected in a clean environment for finetuning. The average increase on hard mode is 3.0 for ACT and 2.0 for $\pi_0$. These results show that our method is suitable for both the VLA framework, including a pre-trained model (ACT), and the finetuning model ($\pi_0$).

*Table 8.* ACT: Performance (%) comparison of the original simulation data (Ori.) *vs.* augmented data (Aug.) on ACT (Zhao et al., 2023) in Hard vs. Easy scenarios of Robotwin 2.0 (Chen et al., 2025)

| Task Name | Hard | | Easy | |
|---|---|---|---|---|
| | Ori. | Aug. | Ori. | Aug. |
| adjust_bottle | 20.0 | 26.0 (+6.0) | 92.0 | 94.0 (+2.0) |
| beat_block_hammer | 2.0 | 2.0 | 56.0 | 52.0 (-4.0) |
| pick_dual_bottles | 0.0 | 4.0 (+4.0) | 30.0 | 34.0 (+4.0) |
| place_burger_fries | 0.0 | 0.0 | 46.0 | 52.0 (+6.0) |
| open_laptop | 0.0 | 0.0 | 54.0 | 58.0 (+4.0) |
| move_can_pot | 2.0 | 4.0 (+2.0) | 20.0 | 28.0 (+8.0) |
| rotate_qrcode | 0.0 | 0.0 | 2.0 | 2.0 |
| grab_roller | 22.0 | 32.0 (+10.0) | 94.0 | 90.0 (-4.0) |
| average | 6.0 | 9.0 (+3.0) | 49.0 | 51.0 (+2.0) |

*Table 9.* $\pi_0$: Performance (%) comparison of the original simulation data (Ori.) *vs.* augmented data (Aug.) on $\pi0$ (Black et al., 2024) in Hard vs. Easy scenarios of Robotwin 2.0 (Chen et al., 2025)

| Task Name | Hard | | Easy | |
|---|---|---|---|---|
| | Ori. | Aug. | Ori. | Aug. |
| adjust_bottle | 74.0 | 78.0 (+4.0) | 90.0 | 92.0 (+2.0) |
| beat_block_hammer | 20.0 | 16.0 (-4.0) | 42.0 | 48.0 (-6.0) |
| pick_dual_bottles | 12.0 | 10.0 (-2.0) | 56.0 | 60.0 (+4.0) |
| place_burger_fries | 4.0 | 10.0 (+6.0) | 80.0 | 76.0 (+6.0) |
| open_laptop | 46.0 | 54.0 (+8.0) | 84.0 | 84.0 |
| move_can_pot | 2.0 | 4.0 (+2.0) | 56.0 | 64.0 (+8.0) |
| rotate_qrcode | 16.0 | 20.0 (+4.0) | 66.0 | 68.0 (+2.0) |
| grab_roller | 78.0 | 76.0 (-2.0) | 84.0 | 96.0 (+12.0) |
| average | 31.5 | 33.5 (+2.0) | 69.8 | 73.5 (+3.7) |

## A.2. Experiments on LIBERO and LIBERO-Plus

As showed in Table 10, we also evaluate on LIBERO using $\pi_0$ and $\pi_{0.5}$ and observe a slight performance drop when training with augmented data: $\pi_0$ decreases by an average of 0.2 and $\pi_{0.5}$ by 0.5 across four task suites. This slight drop is caused by the evaluation environment of LIBERO, which is almost identical to the original training data, making aggressive augmentation introduce distributional disturbances. What 's more, with the performance of $\pi_0$ and $\pi_{0.5}$ approach full scores, it can hardly evaluate improvement.

**Comparison of mixture and replacement strategies on LIBERO-Plus.** In Table 11, by comparing the use of mixture and replacement on both $\pi_0$ and $\pi_{0.5}$, it is proved that using both strategies can help to boost models' performance on the evaluation benchmark. However, whether to choose the mixture strategy or the replacement strategy depends on the models' features. For example, using the mixture strategy on $\pi_0$ can surpass using the replacement strategy.

*Table 10.* Performance of $\pi_0$ (mixture strategy) and $\pi_{0.5}$(replacement strategy) on LIBERO.

| Task Name | $\pi_0$ | | $\pi_{0.5}$ | |
|---|---|---|---|---|
| | Ori. | Aug. | Ori. | Aug. |
| libero spatial | 96.8 | 96.4 | 98.8 | 97.8 |
| libero object | 98.8 | 98.5 | 98.2 | 98.4 |
| libero goal | 95.8 | 96.1 | 98.0 | 98.2 |
| libero 10 | 85.2 | 84.7 | 92.4 | 91.0 |
| average | 94.1 | 93.9 | 96.8 | 96.3 |

*Table 11.* Performance (%) comparison on spatial suite from LIBERO-Plus using $\pi_0$ and $\pi_{0.5}$, "Rep" means using the Replacement strategy, "Mix" means using the mixture strategy

| Task Name | $\pi_0$ | | | $\pi_{0.5}$ | | |
|---|---|---|---|---|---|---|
| | Ori. | Aug. (Rep) | Aug. (Mix) | Ori. | Aug. (Rep) | Aug. (Mix) |
| light conditions | 75.0 | 75.7 | 78.7 | 94.5 | 97.9 | 99.3 |
| objects layout | 69.6 | 66.2 | 86.2 | 97.9 | 97.4 | 97.1 |
| background textures | 81.1 | 83.4 | 87.6 | 95.7 | 95.3 | 95.3 |
| sensor noise | 19.9 | 16.2 | 18.2 | 91.2 | 93.4 | 93.4 |
| language instructions | 37.9 | 43.2 | 55.9 | 90.0 | 90.0 | 89.1 |
| robot initial states | 10.3 | 8.1 | 6.3 | 82.6 | 84.9 | 83.2 |
| camera view points | 21.3 | 17.4 | 15.2 | 79.3 | 79.5 | 79.3 |
| average | 42.7 | 43.3 | 47.8 | 89.8 | 90.8 | 90.2 |

## A.3. Computation analysis

During the full procedure for transferring video, the computation costs are computed based on setting the pixels of the input video to 960*720 (width, height). We use one A800 80GB GPU for generating, the average time saving percentage (Acceleration Rate) is shown in Figure 8. The average time on the selected ten tasks to augment one video (episode) is 560 seconds on one GPU. Augmenting the full dataset of 50 videos requires 7.7 GPU hours. Training the RDT-1B model on 50 episodes requires 56 GPU hours; augmentation adds only 13.8% overhead.

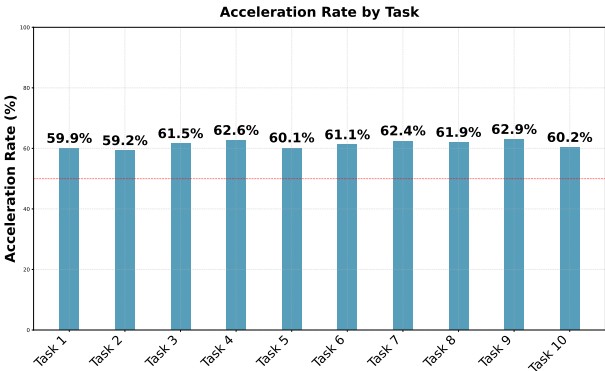

*Figure 8.* Acceleration rates across 10 Robotwin 2.0 tasks. Our method reduces runtime by over 60% on average. Task 1-10: beat block hammer, adjust bottle, handover block, haing mug, pick dual bottles, place a2b right, place burger fries, place dual shoes, stack blocks two, pick diverse bottles.

## A.4. Justification on hyperparameters

The choice of k=0.4 and a=8 when transferring videos is based on a trade-off between efficiency and visual–geometric fidelity, validated through ablations on a representative task: Adjust Bottle. We vary k from 0.2 to 0.8 while keeping a=8 constant. As shown in Table 12, increasing k generally leads to a slight degradation in geometric fidelity (e.g., RMSE and

Abs.Rel increase), while time cost decreases marginally. However, the overall performance remains stable.

*Table 12.* Varying k (with a=8)

| $k$ | RMSE | Abs.Rel | Sq.Rel | sim | time cost |
|-----|------|---------|--------|-------|-----------|
| 0.2 | 0.26 | 0.14 | 0.06 | 25.04 | 467s |
| 0.4 | 0.28 | 0.16 | 0.07 | 26.37 | 441s |
| 0.6 | 0.24 | 0.13 | 0.04 | 25.17 | 446s |
| 0.8 | 0.32 | 0.18 | 0.08 | 25.16 | 425s |

We then fix k=0.4 and vary a from 4 to 12. As reported in Table 13, the configuration a=8 achieves the best balance: it maintains high semantic alignment with stable geometry (lowest Sq.Rel and competitive RMSE/Abs.Rel), while achieving the lowest time cost (441 s). Smaller values of a lead to higher time costs without significant accuracy gains, while larger values of a degrade geometric fidelity.

*Table 13.* Varying a (with k=0.4)

| $a$ | RMSE | Abs.Rel | Sq.Rel | sim | time cost |
|-----|------|---------|--------|-------|-----------|
| 4 | 0.26 | 0.16 | 0.06 | 25.47 | 501s |
| 6 | 0.24 | 0.15 | 0.06 | 25.14 | 446s |
| 8 | 0.28 | 0.16 | 0.07 | 26.37 | 441s |
| 12 | 0.26 | 0.14 | 0.05 | 25.36 | 395s |

We further validate robustness on real-world tasks using the same parameters. The results in Table 14 show that the chosen configuration generalizes well across both simulated and real-world scenarios.

*Table 14.* Real-world Tasks Validation

| Task | RMSE | Abs.Rel | Sq.Rel | sim |
|------|------|---------|--------|-------|
| slot pen into holder | 0.156 | 0.150 | 0.02 | 27.04 |
| stack pipe | 0.20 | 0.14 | 0.03 | 26.8 |

## B. Implementation Details

### B.1. Using details of cosmos-transfer2.5

We use the original video as input, the depth video obtained from depth anything, and the prompt generated by our method as the control condition. The input pixels is set to be 960*720 (height, width). The total diffusion steps are set to 35. The number of frames per chunk is set to be 93.

### B.2. LLM Rewriter

The designed prompt we are using to ask Qwen3-8B to rewrite the conditional prompt is as follows, where "video_description" means video caption, "task_instruction" means instruction got from the original dataset.

```
<user>: You are given a video caption describing a robot manipulation scene and a
brief origin video description. Your task is to generate a new video caption for a
text-to-video generation model. The new captions should:
* Only change the table surface or the object's surface, like changing the material to
create differences.
* Output style is expected to look like the following strictly: The video is a
demonstration of robotic manipulation, likely in a laboratory or industrial setting.
It features a single robotic arm interacting with a plastic bottle. The setting is
a room with a polished stainless steel countertop, which reflects overhead lights
and provides a sterile, metallic backdrop for the activity. The robotic arm, marked
```

```
'AGILE X', is positioned above the bottle, which is filled with a dark liquid.  At the
beginning, the bottle is standing upright on the counter.  The robotic arm approaches
the bottle, its gripper maneuvering with precision as it positions itself.  The
arm's gripper then grasps the bottle firmly by its neck.  As the arm lifts the bottle
smoothly, the liquid inside sways gently.  The entire process highlights the precision
and control of the robotic arm.  The camera remains static throughout, focusing on the
interaction between the robotic arm and the bottle, allowing viewers to observe the
detailed movements involved in the task.
* The central focus of the caption should be on a table's surface.  The table should be
made of wood, stainless steel, or marble.
* Output is expected to be brief and easy enough for the diffusion model to understand.
* The final output should contain only the new caption with no additional commentary or
explanation.  The video caption is:  {video_description}
* The brief origin video description from instruction is:  {task_instruction}, make
sure that the output content has the same meaning and object name as the given brief
description.

<example output 1>:  The video is a demonstration of robotic manipulation, likely in
a laboratory or industrial setting.  It features a single robotic arm interacting with
a green plastic bottle.  The setting is a room with a polished wooden table, which
provides a warm, natural backdrop for the activity.  The robotic arm, marked 'AGILE',
is positioned above the bottle.  At the beginning, the bottle is standing upright
on the table.  The robotic arm approaches the bottle, its gripper maneuvering with
precision as it positions itself.  The arm's gripper then grasps the bottle firmly by
its neck.  As the arm lifts the bottle smoothly, the process highlights the precision
and control of the robotic arm.  The camera remains static throughout, focusing on the
interaction between the robotic arm and the bottle, allowing viewers to observe the
detailed movements involved in the task.
<example output 2>:  The video is a demonstration of robotic manipulation, likely in
a laboratory or industrial setting.  It features a single robotic arm interacting
with a metal kitchen pot.  The setting is a room with a polished wooden countertop,
which contrasts with the metallic elements and provides a warm, natural backdrop for
the activity.  The robotic arm, marked 'AGILE X', is positioned above the pot, which
is empty and neatly placed.  At the beginning, the kitchen pot is resting on the
counter.  The robotic arm approaches the pot, its gripper maneuvering with precision
as it positions itself.  The arm's gripper then grasps the pot firmly by its handles.
As the arm lifts the pot smoothly, the action demonstrates meticulous control.  The
entire process showcases the precision and dexterity of the robotic arm.  The camera
remains static throughout, focusing on the interaction between the robotic arm and the
pot, allowing viewers to observe the detailed movements involved in the task.
<example output 3>:  The video is set in an experimental lab, featuring a robotic arm
labeled 'AGILE'. The arm hovers above a white sneaker on a blue mat, which rests on
an elegant wooden table.  Surrounding the sneaker, various items such as a trophy, a
can, and a small bell are visible.  The robotic arm approaches the sneaker, implying
an interaction, possibly for cleaning or maintenance.
<example output 4>:  The video is a demonstration of robotic manipulation, likely in
a laboratory or industrial setting.  It features a single robotic arm interacting with
a slanted rectangular soap.  The setting is a room with a polished marble countertop,
which beautifully reflects overhead lights and provides an elegant, smooth backdrop
for the activity.  The robotic arm, marked 'AGILE X', is positioned above a light wood
cabinet.  At the beginning, the cabinet drawer is closed.  The robotic arm approaches,
its gripper maneuvering with precision as it opens the drawer.  The arm's gripper then
gently grasps the soap, noted for its slight groove pattern, and places it inside the
open drawer.  The entire process highlights the precision and control of the robotic
arm.  The camera remains static throughout, focusing on the interaction between
the robotic arm, the soap, and the drawer, allowing viewers to observe the detailed
movements involved in the task.
```

### B.3. Velocity Caching Strategy

Algorithm 1 provides pseudocode for our velocity caching mechanism. In practice, we set $k = 0.4$, $\alpha = 8$, and $m = 3$ based on empirical tuning.

---

**Algorithm 1** Three-Stage Velocity Caching

---

**Require:** Total steps $N$, stable threshold $k$, cache interval $\alpha$, adjustment steps $m$
1: **for** $t = 0 \rightarrow N - 1$ **do**
2:    Determine current phase (Initial, Stable, Adjustment)
3:    **if** Stable Phase and $t \mod \alpha \neq 0$ **then**
4:       $v_t \leftarrow v_{\text{cache}}$                                              {Reuse previous value}
5:    **else**
6:       $v_t \leftarrow v_\theta(x_t, t)$                                                  {Recompute}
7:       Update $v_{\text{cache}} \leftarrow v_t$
8:    **end if**
9:    $x_{t+1} \leftarrow x_t + \Delta t \cdot v_t$
10: **end for**

---

## B.4. Real-World Experiments

**Experimental setup.** We deploy our method on an AgileX Piper manipulator equipped with a wrist-mounted camera for visual observation. Data collection is performed via teleoperation using a master-follower arm configuration. We evaluate on two representative manipulation tasks: (1) *Stack Tape*: "stacking a brown tape roll on top of another brown tape"; and (2) *Slot Pen*: "inserting a yellow pen into a holder." For each task, we collect approximately 80 teleoperated demonstrations (80 episodes for Stack Tape, 77 episodes for Slot Pen). Using our proposed video augmentation pipeline, we generate an equal number of augmented trajectories at a 1:1 ratio, resulting in 160 and 154 total training episodes, respectively.

**Distribution shift settings.** To systematically evaluate generalization capability, we design three test conditions with increasing levels of distribution shift:

*(1) In-Distribution (I.D.).* Target objects (tape or pen) are randomly displaced within a 5cm range from training positions, using the original white table background.

*(2) Position Shift (OOD).* Target objects are displaced beyond the 5cm training range, introducing spatial perturbations while maintaining the standard white background.

*(3) Background Shift (OOD).* Objects remain within the 5cm training range, but the table background is changed from white to black to introduce a visual domain shift. These three scenarios are visualized in Figure 5.

**Evaluation protocol.** Each task is evaluated over 10 trials per test condition. A trial is considered successful if the robot completes the full manipulation sequence: for Stack Tape, the robot must grasp and place the tape stably on top of the target tape; for Slot Pen, the pen must be fully inserted into the holder slot. We compare models fine-tuned on original data (baseline) against models fine-tuned on augmented data (ours) using the official OpenPi framework with default full-parameter fine-tuning settings on an NVIDIA H200 GPU.

## C. Training configurations

We provide detailed training configurations below:

**Pi0 for LIBERO and LIBERO-Plus in experiment 4.2** followed the standard full fine-tuning configurations provided by the OpenPi repository, which was trained for 30,000 steps with a global batch size of 32 .

**Pi0.5 for LIBERO and LIBERO-Plus in experiment 4.2** followed the standard full fine-tuning configurations provided by the OpenPi repository, which was trained for 30,000 steps with a global batch size of 256 .

**RDT for RoboTwin 2.0 multi-tasks training in experiment 4.1** followed the standard training configurations provided by the RoboTwin 2.0 repository, which was trained for 100,000 steps with a batch size of 16 per GPU on 8 GPUs.

**RDT for RoboTwin 2.0 single-task training in experiment 4.1** followed the standard training configurations provided by the RoboTwin 2.0 repository, which was trained for 10,000 steps with a batch size of 16 per GPU on 4 GPUs.

**Pi0 for RoboTwin 2.0 single-task training in experiment 4.1** followed the standard training configurations provided by the RoboTwin 2.0 repository, which was performed for 30,000 steps using the global batch size of 32.

**ACT for RoboTwin 2.0 single-task training in experiment 4.1** followed the standard training configurations provided by the RoboTwin 2.0 repository, which was trained under a unified setup with a chunk size of 50, batch size of 8, and single-GPU training for 6,000 epochs.

## D. Visualization of Augmented Videos

### D.1. Augmented Video *vs.* Original Video

In Figure 9, we visualize the transferred videos along with their original videos obtained from Robotwin 2.0's domain randomized tasks, including lift pot and pick bottle.

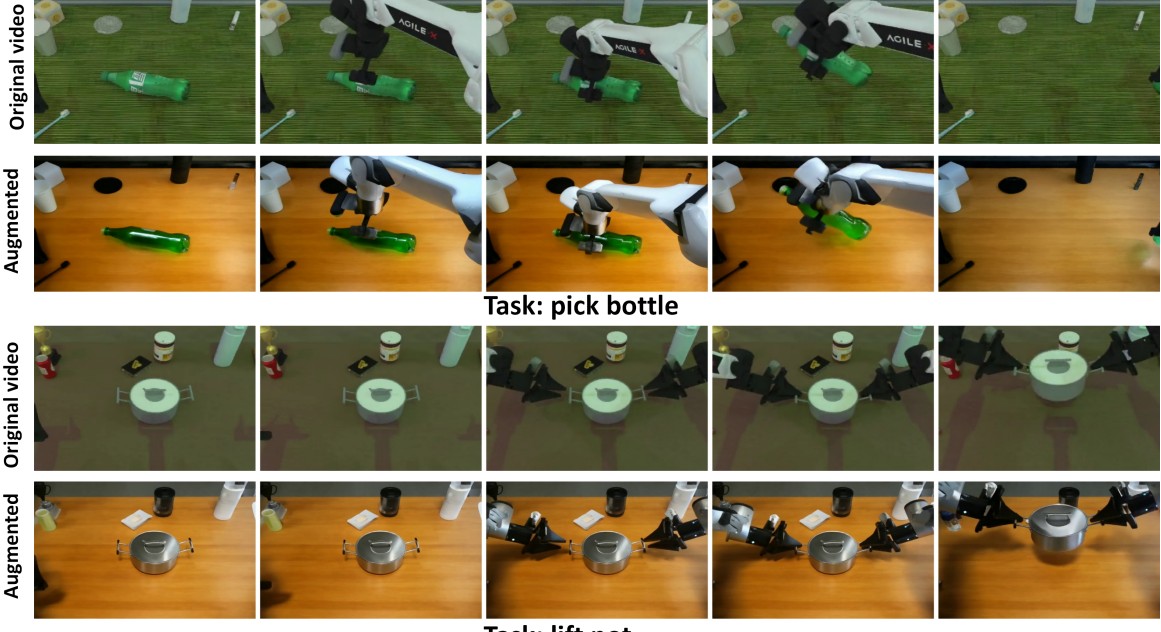

*Figure 9.* Visualizations of augmented videos and origin videos from Robotwin 2.0

In Figure 10, we visualize the transferred videos along with their original videos obtained from the LIBERO official dataset.

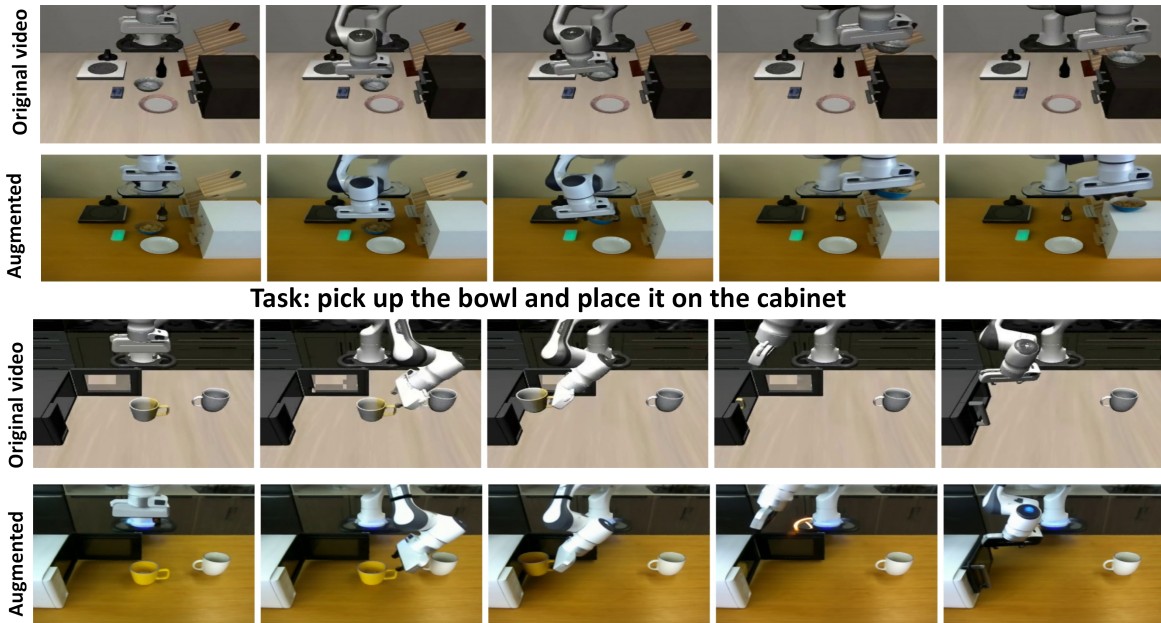

*Figure 10.* Visualizations of augmented videos and origin videos from LIBERO

In Figure 11, we visualize the transferred videos along with their original videos, which were obtained from real-world recordings, including tasks "pick and place carrot" and "remove lid from pot".

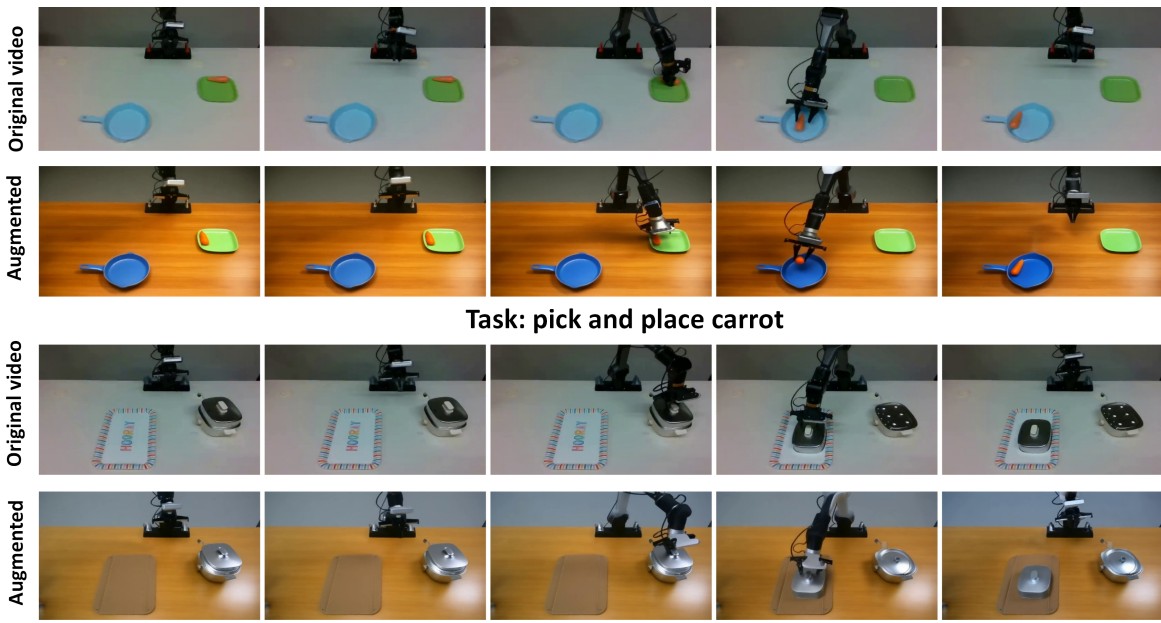

*Figure 11.* Visualizations of augmented videos and original videos from a real robot experiment

In Figure 12, we visualize the transferred videos along with their original videos got from real-world recordings, including tasks "slot pen into holder" and "stack tapes".

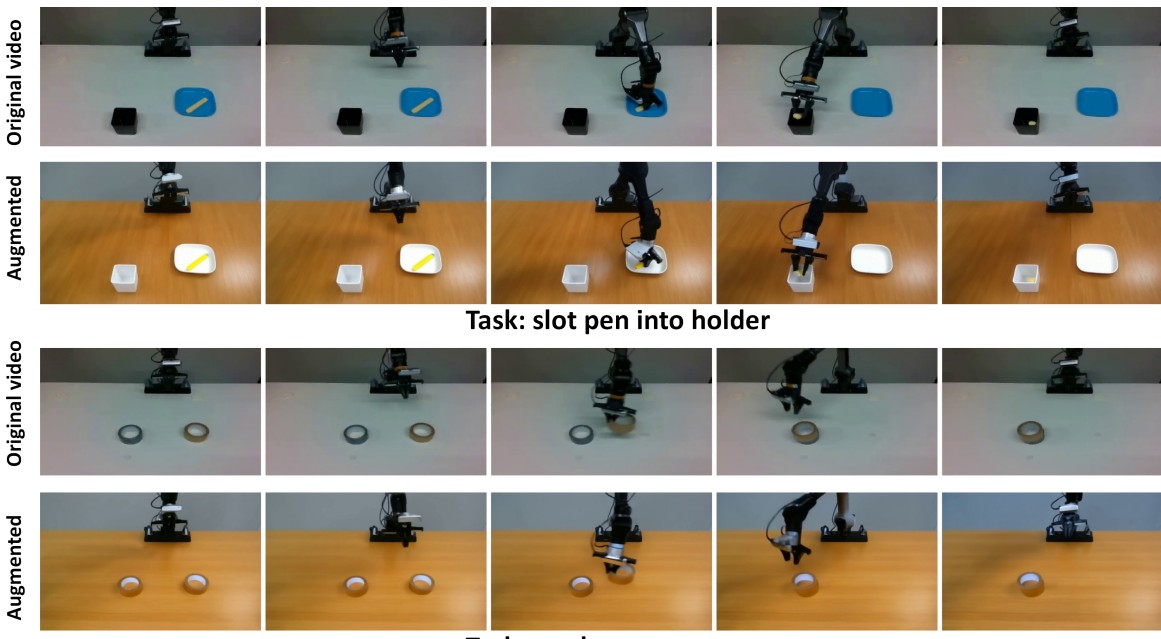

*Figure 12.* Visualizations of augmented videos and original videos from a real robot experiment

## D.2. Augmenting Videos with or without Velocity Caching

In Figure 13, we visualize the transferred videos along with their original videos obtained from Robotwin 2.0's domain clean tasks and compare between using acceleration and not using acceleration.

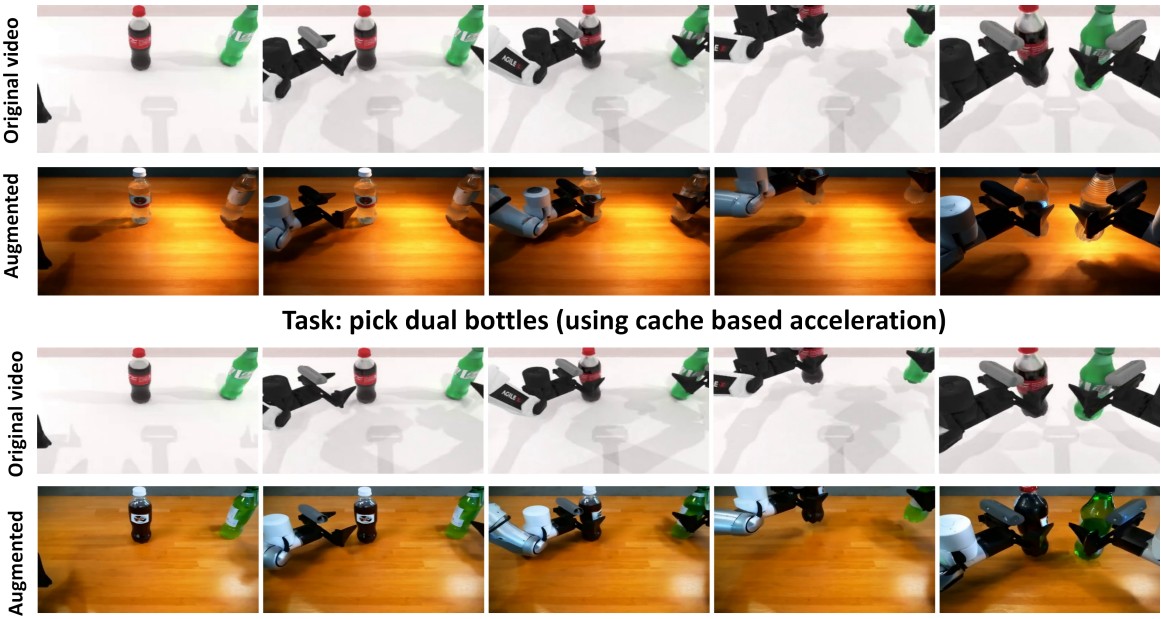

*Figure 13.* Comparing between using cache-based acceleration and not using

In Figure 14, we visualize the transferred videos along with their original videos obtained from Robotwin 2.0 's domain clean tasks and compare using acceleration and not using acceleration.

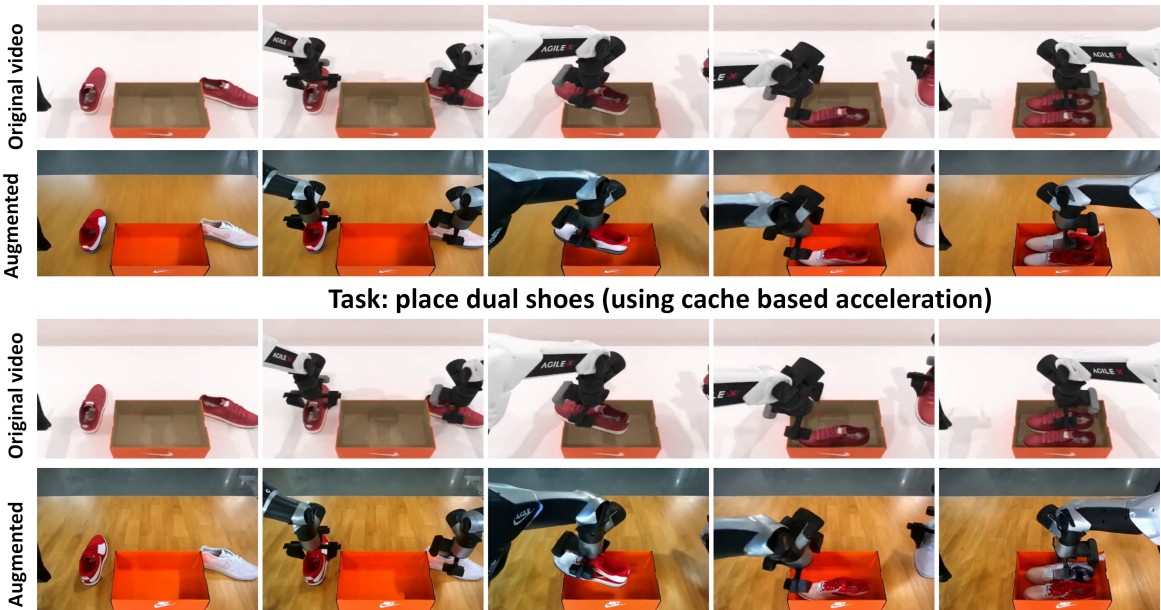

*Figure 14.* Comparing between using cache-based acceleration and not using

