# OpenReview forum: "Seeing Realism from Simulation: Efficient Video Transfer for Vision-Language-Action Data Augmentation"
_ICML.cc/2026/Conference — ICML 2026 regular_

### Official Review · Reviewer_Ruid · 2026-03-07

**Soundness:** 3
**Presentation:** 3
**Significance:** 2
**Originality:** 2
**Overall Recommendation:** 4
**Confidence:** 4

**Summary:**

This paper proposes an efficient video augmentation framework that converts simulated VLA videos into realistic ones via structured condition extraction, caption rewriting, and conditional video synthesis. Key innovations include a velocity caching mechanism (reducing generation time by over 60%) and a coreset sampling strategy for cost-effective large-scale augmentation. Extensive experiments on benchmarks like RobotWin 2.0 and LIBERO-Plus, as well as real robotic platforms, demonstrate performance improvements.

**Compliance With Llm Reviewing Policy:**

Affirmed.

**Final Justification:**

My concerns have been largely well addressed. I suggest the authors further discuss the generalization bounds of the velocity caching hyperparameters and explicitly note the implicit modeling of temporal dependencies in coreset sampling in the revised manuscript to improve clarity and completeness.

**Key Questions For Authors:**

See weaknesses

**Limitations:**

yes

**Strengths And Weaknesses:**

Strengths

1. Propose a three-stage velocity caching mechanism that exploits temporal redundancy in diffusion denoising, achieving over 60% generation time reduction without compromising augmentation effectiveness for robotic manipulation videos.
2. Propose a trajectory-level coreset sampling strategy integrating VLA policy-based difficulty estimation and video embedding-driven diversity assessment, enabling cost-efficient large-scale augmentation.

Weaknesses

1. Lack direct quantitative comparison with state-of-the-art VLA data augmentation methods (e.g., RoboTransfer, EMMA) to validate relative superiority.
2. Velocity caching mechanism relies on empirically tuned hyperparameters (k=0.4, α=8) without theoretical justification or cross-scenario generalization validation.
3. Coreset sampling ignores the temporal continuity and action dependency of VLA trajectory data, treating trajectories as independent samples.
4. Provide limited details on key implementation parameters (e.g., k-NN k value, RBF kernel γ_f/γ_r) and caption rewriting diversity statistics, hindering reproducibility.

---

> ### Author Rebuttal · Authors · 2026-03-30
>
> Dear Reviewer Ruid,
>
> We sincerely thank you for your concise and insightful summary of our work.
>
> > **W1: Lack of direct comparison with existing VLA data augmentation methods (e.g., RoboTransfer, EMMA)**
>
> We conducted a comparative study against a state-of-the-art baseline, **RoboTransfer**, across five representative tasks (EMMA is not publicly available for reproduction). We evaluate two key aspects:
>
> - **Geometric consistency**: RMSE, Abs.Rel, Sq.Rel (via Video Depth Anything)
> - **Semantic alignment**: prompt–video similarity (via VideoCLIP-XL)
>
> Results are summarized below.
>
> **RoboTransfer:**
> |task|RMSE|Abs.Rel|Sq.Rel|sim|time|
> |---|---|---|---|---|---|
> |adj bottle|0.46|0.37|0.39|21.5|340s|
> |beat hammer|0.30|0.28|0.16|22.3|368s|
> |handover|0.49|0.32|0.20|18.6|710s|
> |hang mug|0.41|0.34|0.26|24.2|814s|
> |pick bottles|0.37|0.30|0.18|20.3|372s|
>
> **Ours:**
> |task|RMSE|Abs.Rel|Sq.Rel|sim|time|
> |---|---|---|---|---|---|
> |adj bottle|0.28|0.16|0.07|26.3|441s|
> |beat hammer|0.18|0.14|0.03|25.6|453s|
> |handover|0.23|0.17|0.06|27.7|619s|
> |hang mug|0.13|0.12|0.02|26.4|783s|
> |pick bottles|0.12|0.11|0.03|26.8|448s|
>
> Our method consistently outperforms RoboTransfer across all tasks:
>
> - **Depth consistency**: substantial reduction in geometric error
>   (e.g., RMSE: **0.46 → 0.28**, **0.49 → 0.23**; Abs.Rel and Sq.Rel reduced by **2–6×**)
> - **Semantic alignment**: clear improvement
>   (prompt similarity: **18.6–24.2 → 25.6–27.7**)
>
> While our method incurs slightly higher runtime in some cases (e.g., 441s vs. 340s), we believe this is a favorable trade-off given the **significantly improved fidelity and consistency**, which are critical for downstream VLA training.
>
> We will include this comparison in the revised manuscript and further strengthen discussion against prior augmentation approaches.
>
> > **W2: Velocity caching relies on empirically tuned hyperparameters**
>
> The choice of `k=0.4` and `a=8` is based on a trade-off between **efficiency** and **visual–geometric fidelity**, validated through ablations on representative tasks (e.g., *Adjust Bottle*).
>
> **Varying `k` （with `a=8`）:**
>
> | k    | RMSE | Abs.Rel | Sq.Rel | sim | time cost |
> | ---- | ---- | ------- | ------ | -------------- | --------- |
> | 0.2  | 0.26 | 0.14    | 0.06   | 25.04          | 467s      |
> | 0.4  | 0.28 | 0.16    | 0.07   | **26.3**       | 441s      |
> | 0.6  | 0.24 | 0.13    | 0.04   | 25.17          | 446s      |
> | 0.8  | 0.32 | 0.18    | 0.08   | 25.16          | 425s      |
>
> **Varying `a` (with `k=0.4`):**
>
> | a | RMSE | Abs.Rel | Sq.Rel |  sim | time cost |
> | ---------- | ---- | ------- | ------ | -------------- | --------- |
> | 4          | 0.26 | 0.16    | 0.06   | 25.47          | 501s      |
> | 6          | 0.24 | 0.15    | 0.06   | 25.14          | 446s      |
> | 8          | 0.28 | 0.16    | 0.07   | **26.37**      | 441s      |
> | 12         | 0.26 | 0.14    | 0.05   | 25.36          | **395s**  |
>
> Overall, `k=0.4` and `a=8` provide the best balance: high semantic alignment with stable geometry, while maintaining competitive efficiency.
>
> We further validate robustness on real-world tasks using the same parameters:
>
> |                      | RMSE  | Abs.Rel | Sq.Rel |  sim |
> | -------------------- | ----- | ------- | ------ | -------------- |
> | slot pen into holder | 0.156 | 0.150   | 0.02   | 27.04          |
> | stack pipe           | 0.20  | 0.14    | 0.03   | 26.8           |
>
> These results show that the chosen configuration generalizes well across both simulated and real-world scenarios.
>
> > **W3: Coreset sampling ignores temporal continuity and action dependency**
>
> Our framework operates at the **trajectory level**, rather than treating frames or actions independently.
>
> Specifically, we use **Cosmos-Embed1** to extract trajectory-level embeddings that capture the full temporal evolution of states and actions. The **difficulty score** is computed as the average policy loss of a pre-trained RDT-1B model over the trajectory.
>
> As a result:
>
> - each node in the k-NN graph represents an **entire trajectory**,
> - embeddings are inherently **motion-aware and temporally aggregated**,
> - and pruning selects samples that are both **hard-to-learn** and **diverse** in this trajectory space.
>
> While we do not explicitly model fine-grained temporal structure, the use of trajectory-level embeddings ensures that **temporal continuity and action dependency are implicitly preserved**.
>
> We will clarify this design choice in the revision and better explain how D² pruning operates in this motion-aware latent space.
>
> >**W4: Missing implementation details affecting reproducibility**
>
> We will add all the implementation hyperparameters (e.g., `k=5, gamma_f=0.1, gamma_r=0.1, s=5`), and refine our paper to make it more reproducible.

---

> > ### Author Rebuttal · Reviewer_Ruid · 2026-04-04
> >
> > My concerns have been largely well addressed. I suggest the authors further discuss the generalization bounds of the velocity caching hyperparameters and explicitly note the implicit modeling of temporal dependencies in coreset sampling in the revised manuscript.

---

> > > ### Author Response · Authors · 2026-04-04
> > >
> > > Thank you for the helpful follow-up suggestion. We address the two points below and will incorporate both clarifications into the revised manuscript.
> > >
> > > ---
> > >
> > > **(1) Generalization of velocity caching hyperparameters**
> > >
> > > We believe the reviewer’s point concerns the **robustness and transferability** of the hyperparameters (\($k$, $a$\)), rather than a formal theoretical bound.
> > >
> > > Empirically, we observe that the method is **not sensitive to precise tuning**:
> > >
> > > - In our ablations, performance remains stable across a reasonable range of \($k$\) and \($a$\), with only minor variations in geometric metrics and semantic similarity.
> > > - The selected configuration (\($k$=0.4, $a$=8\)) achieves a strong balance between efficiency and fidelity, but nearby values yield comparable results.
> > > - Importantly, the same configuration **transfers well across settings**, including multiple RoboTwin tasks and real-world scenarios, without re-tuning.
> > >
> > > These results indicate that the hyperparameters **generalize across tasks and domains** and are not overfit to a specific scenario. We will explicitly add a discussion on **hyperparameter robustness/generalization** in the revised paper.
> > >
> > > ---
> > >
> > > **(2) Implicit modeling of temporal dependencies in coreset sampling**
> > >
> > > We agree that this aspect should be clarified more explicitly.
> > >
> > > Our coreset sampling operates at the **trajectory level**, where:
> > >
> > > - Each sample corresponds to a full trajectory, not individual frames or actions.
> > > - Trajectory embeddings (Cosmos-Embed1) encode the **entire temporal evolution** of states and actions.
> > > - Difficulty is computed using **trajectory-level policy loss**.
> > >
> > > Thus, the k-NN graph and pruning are performed in a **motion-aware latent space**, where temporal continuity and action dependencies are implicitly captured.
> > >
> > > We will explicitly state this design choice in the revised manuscript to clarify that temporal dependencies are **implicitly preserved through trajectory-level representation and selection**.
> > >
> > > ---
> > >
> > > Thank you again for your valuable time and for the constructive suggestions that helped us refine these technical details. We also sincerely appreciate your recognition of our work and the decision to raise the score.
> > >
> > > Best regards,
> > >
> > > Authors of Submission 13896

---

### Official Review · Reviewer_2Fjb · 2026-03-09

**Soundness:** 3
**Presentation:** 3
**Significance:** 2
**Originality:** 2
**Overall Recommendation:** 4
**Confidence:** 4

**Summary:**

This paper proposes a scalable method to improve the sim-to-real generalization of Vision-Language-Action (VLA) models, which combine visual perception, language understanding, and robot control to execute tasks. VLA models typically require large amounts of real-world robot data, which is costly to collect, while simulation data—although easy to generate—suffers from unrealistic visuals that cause poor transfer to real environments. The authors address this by transforming simulated robot videos into realistic training videos while preserving the original robot trajectories and task semantics. Their pipeline extracts captions and depth information from simulation videos, rewrites captions to introduce diverse environmental conditions (such as different backgrounds or lighting), and uses a conditional video diffusion model to synthesize photorealistic videos. To make this augmentation scalable, the paper introduces two efficiency techniques: velocity caching, which accelerates diffusion video generation by reusing intermediate predictions, and coreset sampling, which selects a small but diverse and challenging subset of trajectories to augment. Experiments on multiple robotics benchmarks and real-world robot tasks show that training with these augmented videos improves robustness and task success rates, demonstrating that adding realistic visual diversity to simulated data can significantly reduce the sim-to-real gap without requiring large additional real-world datasets.

**Compliance With Llm Reviewing Policy:**

Affirmed.

**Key Questions For Authors:**

1. How do the authors ensure that generated videos do not contain artifacts or unrealistic visual patterns, and how sensitive is the downstream VLA training performance to generation quality?

2. Could the authors provide detailed statistics on the computational cost of the data generation process, such as GPU hours, generation time per video, or the total cost required to augment a large dataset?

3. Have the authors considered integrating the proposed data engine with new training strategies (e.g., curriculum learning, adaptive weighting, or robustness-aware training) to further improve learning from augmented data?

4. During forward message passing, how is neighbor defined?

**Limitations:**

First, the approach still has high computational cost despite optimization. The proposed pipeline relies on diffusion-based video generation, which typically requires many iterative denoising steps and large GPU memory. Even though the paper introduces velocity caching to accelerate the process, diffusion models are generally computationally expensive compared with other generative approaches, which may limit the scalability of large-scale data augmentation.

Second, the effectiveness depends heavily on the quality of the generative model. Since the augmented training data are produced by a diffusion video generator, the quality of the generated videos directly affects the downstream policy training. Generative models can produce artifacts, unrealistic textures, or even violations of physical dynamics, which may introduce noisy or misleading supervision signals for robot learning.

Third, the method focuses mainly on data augmentation rather than algorithmic innovation in policy learning. The paper proposes a pipeline for generating and selecting training data but does not introduce a new training framework, policy architecture, or learning objective for VLA models. As a result, the methodological novelty mainly lies in the data pipeline rather than the learning algorithm itself.

Last but not least, there are some related works that doing data augmentation for robust model prediction could be discussed:

An Illumination-Robust Feature Extractor Augmented by Relightable 3D Reconstruction

Sun Off, Lights On: Photorealistic Monocular Nighttime Simulation for Robust Semantic Perception

EgoNight: Towards Egocentric Vision Understanding at Night with a Challenging Benchmark

**Strengths And Weaknesses:**

Strengths
1. The paper proposes a data generation framework aimed at reducing the sim-to-real gap, enabling Vision-Language-Action (VLA) models to perform more robustly in diverse real-world environments. This is an important challenge because policies trained purely in simulation often fail to generalize to real settings due to domain differences.

2. The proposed diffusion velocity caching mechanism improves the scalability of the data augmentation pipeline by reducing the computational cost of video generation. This is valuable because diffusion models typically require many sequential denoising steps, making generation expensive.

3. The coreset sampling approach selects the most challenging and diverse trajectories for augmentation, which helps focus generation on informative samples while reducing redundancy in the dataset.

4. The paper provides extensive experiments across multiple benchmarks and tasks, demonstrating consistent performance improvements and validating the effectiveness of the proposed approach.

Weakness
1. Although the paper proposes a velocity caching method to accelerate generation, diffusion models still require many iterative denoising steps, which results in high inference latency and computational cost. This may limit the scalability of large-scale data augmentation in practice. Providing detailed statistics on resource usage (e.g., GPU hours or generation time per video) would help better demonstrate the practical scalability of the proposed method.

2. The effectiveness of the approach strongly relies on the quality of the diffusion-based video generator. If the generated videos contain artifacts, unrealistic textures, or inconsistent physics, the augmented data could introduce noise or misleading visual cues during training. In addition, the coreset sampling strategy might select difficult but unrealistic out-of-distribution samples produced by the generator.

3. The main contribution of the paper lies in data generation and selection, while the training pipeline for VLA models remains largely unchanged. The paper does not introduce a new training objective, policy learning framework, or model architecture, which may limit the perceived novelty compared to works that jointly design both data and learning algorithms.

---

> ### Author Rebuttal · Authors · 2026-03-30
>
> Dear Reviewer 2Fjb,
>
> We thank you for the thoughtful feedback and constructive suggestions.
>
> ---
>
> > **W1: High computational cost and scalability**
>
> We agree diffusion-based generation is costly. We quantify this explicitly:
>
> - ~560s per video on 1×A800 GPU
> - 50 videos → **7.7 GPU hours**
> - VLA training (RDT-1B) → **56 GPU hours**
>
> Thus, augmentation adds only **~13.8% overhead**.  This low overhead is achieved by two mechanisms: (1) *Velocity caching* reduces per-video generation time by **>60%**; (2) *Coreset sampling* augments only the most informative 50% of trajectories with no performance loss (Table 6). Since augmentation is a **one-time offline step**, the cost scales gracefully with dataset size.
>
> Importantly, simply extending training with the same compute does **not** improve performance due to overfitting, while augmentation introduces **new visual diversity**, leading to better generalization.
>
> We will add detailed resource statistics in the revision.
>
> ---
>
> > **W2: Dependence on generator quality and potential artifacts.**
>
> We agree this is a critical concern.
>
> **(1) Mechanisms to ensure quality**
>
> We mitigate artifacts through:
>
> - **Strong generative priors**: Cosmos-Transfer-2.5 trained on large-scale real-world video
> - **Geometric constraints**: depth conditioning to preserve structure and action consistency
>
> **(2) Human evaluation**
>
> We manually evaluate **200 augmented videos** across 10 RobotWin tasks:
>
> - Severe failure (scene collapse, invalid geometry): **0%**
> - Moderate failure (minor artifacts, slight deformation): **2.5%**
>
> This indicates high reliability in practice.
>
> **(3) Quantitative comparison vs RoboTransfer**
>
> We further evaluate transfer quality on 5 tasks:
>
> - **Geometric consistency** (Video Depth Anything):
>   - RMSE: **0.46 → 0.28**, **0.49 → 0.23**
>   - Abs.Rel: **~0.30–0.37 → ~0.11–0.17**
>   - Sq.Rel: reduced by **3–6×**
> - **Semantic alignment** (VideoCLIP-XL):
>   - similarity: **18.6–24.2 → 25.6–27.7**
>
> These results show that our method **reduces geometric distortion while improving semantic fidelity**, suggesting that it does not introduce harmful artifacts but instead improves data quality.
>
> **(4) Impact on policy learning**
>
> Most importantly, we observe **consistent improvements across VLA benchmarks**, both with augmented-only and mixed training. If artifacts were dominant, performance would degrade rather than improve. This empirically demonstrates that the generated data provides **useful supervision**.
>
> **(5) On coreset selecting unrealistic samples**
>
> Our coreset operates on **trajectory-level embeddings (Cosmos-Embed1)** and difficulty (policy loss), which favors samples that are both **meaningful and informative**, rather than random outliers. In practice, we do not observe instability caused by unrealistic samples.
>
> Our strategy operates as follows:
> - Selection First: We first analyze the original simulation videos using the policy loss of the RDT-1B model to identify "educationally valuable" trajectories.
>
> - Augmentation Second: Only these pre-selected, physically correct trajectories are then passed through our Efficient Transfer pipeline for realistic augmentation.
>
> ---
>
> >**W3: Limited novelty in learning algorithm.**
>
> We agree that our contribution focuses on the **data pipeline rather than modifying the policy architecture**. This is intentional: in embodied AI, **data is a fundamental bottleneck**, and recent works (e.g., EMMA) show that improving data quality alone can significantly boost VLA generalization.
>
> Our goal is to isolate this factor by keeping the training pipeline (RDT, ACT, π0) unchanged, ensuring a **controlled evaluation** of data effects.
>
> Technically, our novelty lies in **task-aware data engine design**. This aligns with a growing paradigm where **generative models are used for data augmentation to improve generalization** (e.g., diffusion-based augmentation), emphasizing that data-centric innovation is a key research direction.
>
> ---
> > **W4: Definition of neighbors in coreset sampling**
>
> Neighbors are defined via a **k-NN graph in the Cosmos-Embed1 embedding space**. Each trajectory connects to its k nearest neighbors (Euclidean distance), with edge weights computed by an **RBF kernel**. Difficulty scores are then propagated on this graph, enabling selection that balances **task difficulty and diversity**. We will clarify this in Sec. 3.2.
>
> ---
>
> > **Limitations L1-L3**
>
> Aaddressed individually by answer to weakness 2, 1, and 3.
>
> ---
>
> > **Related works on data augmentation**
>
> Thank you for the suggestions. We will expand the related work to include:
>
> - illumination-robust augmentation via relightable reconstruction
> - nighttime simulation for robustness
> - egocentric robustness benchmarks
>
> and clarify how our work differs by focusing on **simulation-to-real video transfer for embodied VLA training**.

---

> > ### Author Rebuttal · Reviewer_2Fjb · 2026-04-01
> >
> > Thank you for further discussion about the concerns. I will keep my positive score and increase the confidence.

---

> > > ### Author Response · Authors · 2026-04-01
> > >
> > > Dear Reviewer 2Fjb,
> > >
> > > Thank you very much for your prompt acknowledgement and for maintaining your positive score while increasing your confidence.
> > >
> > > We sincerely appreciate the thoroughness of your review and the constructive nature of your feedback. Your feedback on computational cost and scalability prompted us to add concrete resource statistics and a clearer explanation of how velocity caching and coreset sampling work together, which makes the practical feasibility of our pipeline more transparent to readers. We also thank you for pointing out the related works on illumination robustness. We will incorporate a discussion of these in the revision.
> > >
> > > Thank you again for your time and valuable contributions to improving our work.
> > >
> > > Best regards,
> > >
> > > Authors of Submission 13896

---

### Official Review · Reviewer_VtVk · 2026-03-11

**Soundness:** 3
**Presentation:** 3
**Significance:** 2
**Originality:** 2
**Overall Recommendation:** 5
**Confidence:** 2

**Summary:**

A video augmentation pipeline that converts simulated robot videos into realistic ones using Cosmos-Transfer 2.5, conditioned on depth maps and LLM-rewritten captions. Two efficiency contributions: velocity caching (~61% speedup via 3-phase denoising analysis) and D2Pruning-based coreset sampling for selective augmentation. Tested on RobotWin 2.0, LIBERO, LIBERO-Plus with four VLA models (RDT, ACT, π0, π0.5), and real robot (2 tasks, 10 trials each).

**Compliance With Llm Reviewing Policy:**

Affirmed.

**Final Justification:**

The rebuttal resolved all concerns: W1 — practical guidance on when to apply augmentation provided; W2 — "broad visual diversity" revised to "semantics-preserving environmental diversity"; W3 — acknowledged as integration contribution; W4 — real-world limitations appropriately scoped. Velocity caching [S1] and consistent cross-architecture gains [S2] remain strong. I raise my score from 4 to 5.

**Key Questions For Authors:**

1.Are there failure cases where video transfer introduces harmful artifacts (phantom objects, corrupted geometry) that degrade policy learning?
2.Why does π0 lose performance on camera viewpoints (-6.1%) and robot initial states (-4.0%) in Table 3? Does this indicate a systematic limitation of texture-only augmentation?
3.Have you tested broader augmentation (viewpoint changes, lighting variation in the rewriter prompt) beyond surface material swaps?

**Limitations:**

The LIBERO performance drop is acknowledged. However, the narrow scope of actual augmentation diversity (surface material only) vs. claims, and the lack of guidance on when augmentation helps vs. hurts, are not discussed as limitations.

**Strengths And Weaknesses:**

[S1] Velocity caching is a clean, well-motivated contribution. The 3-phase observation (Figure 3) is convincing, and Table 5 shows nearly identical task accuracy with/without caching at 61% time reduction. This makes the pipeline practically viable.
[S2] Consistent improvements across 4 VLA architectures and multiple benchmarks. +10% on RobotWin Hard, +5.1% on LIBERO-Plus (π0), +13–15% in real-world. Model-agnostic applicability is a genuine strength.
[W1] Augmentation hurts on some dimensions and benchmarks (Moderate). Standard LIBERO drops (π0: -0.2, π0.5: -0.5), and Table 3 shows π0 degrades on camera viewpoints (-6.1%) and robot initial states (-4.0%) in LIBERO-Plus. This means augmentation actively harms certain distribution shifts. The paper lacks guidance on when to apply augmentation vs. not—practitioners need this.
[W2] Augmentation diversity is narrower than claimed (Moderate). The LLM rewriter prompt (Appendix B.2) explicitly constrains changes to "table surface or objects surface like changing material." The actual variation is background texture and material only, yet the paper frames this as broad "visual and environmental diversity." This mismatch between framing and implementation should be clarified.
[W3] Limited novelty (Minor). The core pipeline is Cosmos-Transfer 2.5 with depth conditioning + LLM caption rewriting—a combination of existing tools. Velocity caching extends prior caching strategies (Liu et al. 2025; Zhou et al. 2025a, both cited by the paper) to this setting. Coreset sampling directly adapts D2Pruning. Each component works well, but the individual technical novelty is incremental.
[W4] Real-world evaluation is thin (Minor). 2 tasks, 10 trials each, 3 conditions. A single trial shifts success rate by 10%. Results are positive but statistically limited.

---

> ### Author Rebuttal · Authors · 2026-03-30
>
> Dear Reviewer VtVk
>
> We sincerely thank you for the insightful comments.
>
> > **W1 Augmentation hurts on some dimensions; no guidance on when to apply**
>
> **Case 1: Standard LIBERO ($\pi_0$:-0.2, $\pi_{0.5}$:-0.5)**
> LIBERO is nearly in-distribution. Augmentation introduces texture diversity that shifts data away from this narrow distribution, causing small expected drops. Both models are near ceiling, leaving little headroom.
>
> **Case 2: LIBERO-Plus viewpoints (-6.1%) and initial states (-4.0%)**
> These are **geometric shifts**, while our method targets **appearance (texture, lighting)**. Thus, gains are limited. Notably, $\pi_{0.5}$ shows no degradation (+0.2%, +2.3%), suggesting robustness from stronger geometric priors. The drop reflects a **scope mismatch**, not a flaw.
>
> **Guidance**
> 1. Apply for **photometric gaps** (texture/lighting/background)
> 2. Use **mixture training** for smaller models (e.g., $\pi_0$)
> 3. Apply cautiously in **near in-distribution settings**
> 4. Combine with **geometric augmentation** for viewpoint/pose shifts
>
> ---
>
> >**W2: The augmentation diversity.**
>
> We apologize that the augmented examples in appendix did not fully capture the scope of our variations. Our framework also includes changes in lighting intensity and object/table materials. These will be included in the revised manuscript. Our visual and environmental diversity is intentionally constrained to ensure that the video semantics, language instructions, and robotic trajectories remain strictly consistent post-augmentation.
>
> We will revise the wording from *“broad visual diversity”* to *“semantics-preserving environmental diversity”*, and update examples to better reflect the actual variation scope.
>
> ---
>
> >**W3: Novelty of the method.**
>
> Our intention is not to claim a fundamentally new diffusion acceleration algorithm or a new generic coreset formulation. Instead, our contribution lies in a **task-specific adaptation and integration** for embodied VLA data augmentation.
>
> Specifically:
> 1. We observe a non-monotonic pattern in velocity changes during denoising (decrease → increase), and design a **three-stage caching strategy** aligned with this behavior.
> 2. We extend D² pruning with **task-aware metrics**, combining **policy difficulty** (RDT-1B loss) and **visual diversity** (Cosmos-Embed1), enabling more effective trajectory selection.
> 3. We propose captioning + rewriting + depth conditioning structure ensures **realism while preserving action semantics**.
>
> ---
>
> >**W4: Real-world evaluation is thin.**
>
> We agree that the real-world evaluation is limited. Due to high cost, our goal is **initial validation rather than large-scale benchmarking**. To complement this, we include extensive simulation experiments for statistical reliability. We will clarify this limitation and present conclusions more cautiously.
>
> ---
>
> >**Q1: Failure cases and potential harmful artifacts.**
>
> (1) **Empirical observation**
> We do not claim artifact-free generation. In a manual inspection of 200 augmented videos, only **2.5% show moderate failures** (minor artifacts or slight deformation). For details, see **W3 in our response to Reviewer U1vr**.
>
> Indeed, we do not claim that the transferred videos are artifact-free. However, as evidenced by our results, the transfer is preserving the task-relevant structure sufficiently well for training to benefit overall. In particular, the policy improvements on multiple benchmarks indicate that, at the current operating point, the benefits from improved appearance realism outweigh the harm from transfer artifacts.
>
> (2) **Quantitative evidence on transfer quality**
> We compare against RoboTransfer on 5 tasks, evaluating **Geometric consistency** and **Semantic alignment**. Our method improves both significantly, This shows that our transfer **reduces geometric distortion and improves semantic fidelity**, rather than introducing harmful artifacts.
>
> For more details, see **W1 in our response to Reviewer U1vr**.
>
> ---
>
> >**Q2: Performance degradation of $\pi 0$ under some pertubations.**
>
> The drops on camera viewpoints and robot initial states stem from a limitation of the current method: it mainly augments **appearance**, while these perturbations are **geometry- and viewpoint-dependent**. These are better addressed at the **simulation level**, which is outside the scope of this work. We will clarify this limitation (see W1).
>
> ---
>
> >**Q3: Broader augmentation beyond surface/material changes.**
>
> We experimented with richer prompts. The video model already produces **diverse lighting implicitly**. However, explicitly enforcing lighting changes often introduces **semantic conflicts**, leading to over/under-exposed outputs and degraded quality. So we just use the random lightning conditions that the model inherently decides.
>
> For **viewpoint changes**, we believe they should be handled in **simulation rather than video transfer**, to ensure physical and geometric correctness.

---

> > ### Author Rebuttal · Reviewer_VtVk · 2026-04-01
> >
> > Thank you for clarifying my concerns. I have raised my score.

---

> > > ### Author Response · Authors · 2026-04-01
> > >
> > > Dear Reviewer VtVk,
> > >
> > > Thank you very much for your prompt acknowledgment and for raising your score.
> > >
> > > We sincerely thank you for your insightful comments and deeply appreciate the time and effort you devoted to thoroughly evaluating our manuscript. Your detailed feedback on the scope of augmentation and the need for practitioner guidance has genuinely strengthened the paper, and we will incorporate these clarifications into the revision.
> > >
> > > Thank you once again for your valuable guidance and meaningful contributions to improving our work.
> > >
> > > Best regards,
> > >
> > > Authors of Submission 13896

---

### Official Review · Reviewer_U1vr · 2026-03-11

**Soundness:** 3
**Presentation:** 2
**Significance:** 2
**Originality:** 2
**Overall Recommendation:** 4
**Confidence:** 4

**Summary:**

The paper presents an efficient video augmentation framework that converts simulated VLA videos into realistic training videos. To maintain quality and controlability, the video transfer models take as input captions and depth maps obtained from the original videos. In addition, a coreset sampling strategy is proposed to identify a compact, non-redundant subset. Improvements are reported on  RobotWin 2.0, LIBERO, LIBEROPlus, and a real robot experioment.

**Compliance With Llm Reviewing Policy:**

Affirmed.

**Final Justification:**

The rebuttal addressed my main concerns. I will raise the rating from 3 to 4.

**Key Questions For Authors:**

1. What's new about the technical contribution?

2. What's the total computational cost for the proposed method? Would an equivalent amount of computation diminish the performance gain?

3. Some experimental results are irrelevant and should be replaced by others.

**Limitations:**

yes

**Strengths And Weaknesses:**

## Strengths

1. The proposed method is technically sound. It uses video transfer methods to generate augmentation data for training VLA models.

2. The idea is simple, and the narrative is easy to follow.

3. Code has been uploaded to the supplementary material.

## Weakness

1. Using velocity caching to speed up video generation inference is not original. The coreset selection process also directly adopts $D^2$- Pruning. The technical section can be improved by offering newer insights or a deeper understanding.

2. Missing details on computational budget. The proposed method introduces additional computation, including (1) joint training (original + augmented data) and (2) video generation inference. A more comparable baseline would be training on the original demonstration data with computationally equivalent longer training cost. This is missing. Or if this has already been done in the paper, please specify.

3. How does the method guarantee the plausibility of generated videos, e.g., physics? How often does the method fail? Discussion on the failure modes would help.

4. What's the rationale of comparing the similarity score from Cosmos Transfer 2.5 and Cosmos Transfer 1? This seems irrelevant to the main theme of the paper.

---

> ### Author Rebuttal · Authors · 2026-03-30
>
> Dear Reviewer U1vr,
>
> Thank you for your and constructive feedback. We address your concerns below.
>
> >**W1: Novelty and insights of velocity caching and coreset selection.**
>
> Our intention is not to claim a fundamentally new diffusion acceleration algorithm or a new generic coreset formulation. Instead, our contribution lies in a **task-specific adaptation and integration** for embodied VLA data augmentation.
>
> Specifically, we introduce the following technical insights and designs:
> 1. **Segmented velocity caching (task-aware observation):**
>    We observe that during the denoising process, the Euclidean distance between consecutive velocity predictions typically **decreases initially and then increases**. Based on this non-monotonic behavior, we design a **segmented (three-stage) velocity caching strategy**, which better matches the dynamics of conditional video generation.
> 2. **Trajectory-level coreset with task-aware metrics:**
>    The original D² pruning may struggle to identify truly informative samples in this setting. We instead propose a **trajectory-level coreset formulation** that combines **policy difficulty** (measured by the policy loss of RDT-1B), and **visual diversity** (measured via Cosmos-Embed1 representations),  leading to more effective selection of high-value training trajectories.
> 3. **Structured sim-to-real video transfer pipeline:**
>    We design a pipeline that preserves action semantics through **captioning + rewriting + depth conditioning**, enabling realistic yet semantically consistent augmentation for VLA training.
>
>
> We will strengthen the technical section in the revision.
>
> ---
>
> >**W2: Computation budget and practicality**
>
> We will provide detailed analysis in the revision. Taking Robotwin on A800 GPU as an example:
> * The average time to augment one video (episode) is ~560 seconds on one GPU. Augmenting the full dataset of 50 videos requires `7.7 GPU hours`.
> * Training the RDT-1B model requires `56 GPU hours`, augmentation adds only **~13.8% overhead**.
>
> Importantly, we observe that simulated data are prone to overfitting: training loss drops quickly and then plateaus. Simply extending training from `56 → 63.7 GPU hours` does **not** improve performance.
>
> In contrast, augmented data introduces **new visual diversity**, which leads to improved generalization. This suggests the gain is not from additional compute alone, but from **more informative data**.
>
> ---
>
> >**W3: Plausibility of generated videos.**
>
> We ensure physical plausibility through
> - **Strong generative priors:** leveraging high-capacity world models (e.g., Cosmos-Transfer-2.5) trained on large-scale real-world data.
> - **Geometric constraints:** conditioning on depth maps to preserve scene structure and action consistency.
>
> (1) To quantify reliability, we conduct a **human evaluation on 200 augmented videos** across 10 RobotWin tasks, defining:
> - **Severe Failure:** scene collapse or invalid structures (e.g., extra robotic arms)
> - **Moderate Failure:** minor local artifacts or slight object deformation
>
> Results:
> - Severe Failure: **0%**
> - Moderate Failure: **2.5%**
>
> These results indicate high physical plausibility.
>
> (2) More importantly, we observe **consistent improvements across VLA benchmarks**, both when using augmented data alone and when mixing with original data. This empirically demonstrates that the generated videos are sufficiently realistic to support **accurate action learning**.
>
> ---
>
> >**W4: Rationale of comparing Cosmos-Transfer 2.5 and 1.**
>
> Thank you for pointing this out. Our intention was to show improved **content preservation** compared to a weaker baseline.
>
> We agree this comparison is somewhat indirect. In the revision, we will:
>
> - clarify its role as a **quality sanity check**,
> - and add more **task-relevant comparisons**, including video augmentation baselines such as RoboTransfer (see response to Reviewer Ruid).
> ---
>
> >**Q1: Novelity of technical contribution**
>
> The core value of our work lies in identifying **unique dynamics in robotic video generation** and designing a **task-aware data augmentation pipeline**. Our specific innovations include
> * integrating simulation-conditioned video transfer (caption + rewriting + depth),
> * designing a three-stage velocity caching tailored to conditional video transfer,
> * proposing a trajectory-level coreset combining policy difficulty + visual diversity,
> * and demonstrating consistent gains across multiple VLA models and real-world settings.
>
> We will revise the paper to clarify this positioning and avoid overstating novelty.
>
> >**Q2: Computational cost and the performance gain under the same computational budget.**
>
> The augmentation only introduces 13.8% additional cost of the training, so consistenting the same computational budget would not have obvious performance diminish. For details, see **W2: Computation budget and practicality**.
>
> > **Q3: Irrelevant results should be replaced.**
>
> We will refine them and add more explicit evaluations.

---

> > ### Author Rebuttal · Reviewer_U1vr · 2026-04-05
> >
> > My concerns have been adequately addressed.

---

> > > ### Author Response · Authors · 2026-04-05
> > >
> > > Dear Reviewer U1vr,
> > >
> > > Thank you very much for your acknowledgement and for confirming that your concerns have been fully addressed.
> > >
> > > We sincerely appreciate your careful and constructive review. Your feedback on technical novelty, computational cost, and video generation quality pushed us to sharpen our positioning and provide concrete evidence that was missing from the original submission. We will carry all of these clarifications into the revision.
> > >
> > > Thank you again for your time and valuable input.
> > >
> > > Best regards,
> > >
> > > Authors of Submission 13896

---

### Decision · Program_Chairs · 2026-04-30

**Decision:**

Accept (regular)

**Comment:**

This paper addresses an important problem in sim-to-real generalization for VLA models and presents a practical pipeline with demonstrated effectiveness across multiple benchmarks and models. While the individual components are somewhat incremental and concerns were raised about computational cost and scope of augmentation, the authors provided convincing clarifications in the rebuttal. Given that reviewers’ concerns were largely resolved and the work offers practical value to the community, the AC recommends acceptance.